# Antimicrobial Nanomaterials for Food Packaging

**DOI:** 10.3390/antibiotics11060729

**Published:** 2022-05-29

**Authors:** Vasanti Suvarna, Arya Nair, Rashmi Mallya, Tabassum Khan, Abdelwahab Omri

**Affiliations:** 1Department of Pharmaceutical Chemistry & Quality Assurance, SVKM’s Dr. Bhanuben Nanavati College of Pharmacy, Mumbai 400056, India; vasanti.suvarna@bncp.ac.in; 2Department of Quality Assurance, SVKM’s Dr. Bhanuben Nanavati College of Pharmacy, Mumbai 400056, India; arya.h.nair@gmail.com (A.N.); rashmi.mallya@bncp.ac.in (R.M.); 3The Novel Drug & Vaccine Delivery Systems Facility, Department of Chemistry and Biochemistry, Laurentian University, Sudbury, ON P3E 2C6, Canada

**Keywords:** antimicrobial agents, nanomaterials, food packaging, active packaging, smart packaging, edible films

## Abstract

Food packaging plays a key role in offering safe and quality food products to consumers by providing protection and extending shelf life. Food packaging is a multifaceted field based on food science and engineering, microbiology, and chemistry, all of which have contributed significantly to maintaining physicochemical attributes such as color, flavor, moisture content, and texture of foods and their raw materials, in addition to ensuring freedom from oxidation and microbial deterioration. Antimicrobial food packaging systems, in addition to their function as conventional food packaging, are designed to arrest microbial growth on food surfaces, thereby enhancing food stability and quality. Nanomaterials with unique physiochemical and antibacterial properties are widely explored in food packaging as preservatives and antimicrobials, to extend the shelf life of packed food products. Various nanomaterials that are used in food packaging include nanocomposites composing nanoparticles such as silver, copper, gold, titanium dioxide, magnesium oxide, zinc oxide, mesoporous silica and graphene-based inorganic nanoparticles; gelatin; alginate; cellulose; chitosan-based polymeric nanoparticles; lipid nanoparticles; nanoemulsion; nanoliposomes; nanosponges; and nanofibers. Antimicrobial nanomaterial-based packaging systems are fabricated to exhibit greater efficiency against microbial contaminants. Recently, smart food packaging systems indicating the presence of spoilage and pathogenic microorganisms have been investigated by various research groups. The present review summarizes recent updates on various nanomaterials used in the field of food packaging technology, with potential applications as antimicrobial, antioxidant equipped with technology conferring smart functions and mechanisms in food packaging.

## 1. Introduction

More than two hundred human diseases ranging from diarrhea to cancer are caused by food that is contaminated with pathogenic bacteria, viruses, parasites, or chemical compounds, with six hundred million new cases and 420,000 deaths per annum [1]. Food products that are obtained from agricultural sources (e.g., fruits) are heavily contaminated by pathogenic organisms due to a lack of safety practices [2]. Similarly, increased international trade has raised the risk of disease transmission through spoiled food and the associated foodborne diseases they cause. As a result, increased focused efforts towards improvement in food packaging systems are needed to limit the risk of foodborne diseases [3].

Food packaging systems offers various benefits, such as longer shelf life, better handling, and protection from physicochemical damages during storage and transport. Therefore, they exhibit a crucial role in the global food industry. Moreover, consumers are interested in innovative, cost-effective, environmentally friendly, and efficient food packaging materials that provide safe, nutritious and high-quality products. Therefore, currently, various elements such as standardization, advertisement, and the dissemination of information to consumers—in addition to an improvement in food protection, storage, handling and transportation—are major driving forces in food packaging research to achieve intended product quality and shelf-life [4]. Moreover, various innovative packaging systems that comprise active and smart/intelligent packaging materials are widely explored in the food industry. Active packaging (AP) is a modification of traditional packaging that offers protection against the growth of pathogenic microorganisms during food storage. The antimicrobial effect of AP is attributed to its incorporation of antimicrobial (natural or synthetic) agents into the packaging material [3]. However, smart packaging contains indicators that allow customers to identify changes in food quality over time. For example, as the pH rises from 2 to 10, the color of a blueberry film changes from rose to blue-green and blackberry films change from red to dark violet [5].

### 1.1. Packaging Materials

#### 1.1.1. Petroleum-Based Plastic Polymer

Plastics that are manufactured from crude oil or natural gas are known as petroleum-based plastics. They are good candidates for the fabrication of food packaging because of their low cost, ease of processing, light weight, oil and chemical resistance, good gas and water vapor-barrier qualities, and easy reusability and recyclability [6]. High-density polyethylene; low-density polyethylene; polypropylene; polystyrene; polyvinyl chloride; and polyethylene terephthalate are examples of typical petroleum or fossil-based plastics that are used in food packaging. These plastics are mainly non-biodegradable, non-renewable, and non-compostable, posing serious environmental and disposal challenges around the world [7]. They are the most difficult to recycle of all the packaging materials [8].

#### 1.1.2. Biodegradable Polymers

The majority of polymers that are used in the packaging industry are non-biodegradable petroleum-based plastic polymer materials, posing a huge environmental hazard. In recent years, considerable interest in the development of biodegradable polymers from renewable resources to address environmental safety issues has been observed. Microorganisms (e.g., bacteria, fungus) use enzymatic catalysis processes to decompose biodegradable polymers that are placed in bioactive habitats (e.g., landfills) [9]. Non-enzymatic mechanisms such as chemical hydrolysis can also break down polymer chains. The end products of biodegraded polymers often contain carbon dioxide, methane, water, biomass, and other natural substances that are advantageous for balancing greenhouse gas emissions [10]. Thus, developing renewable or sustainable packaging that is composed of biodegradable or edible materials, plant extracts, and nanocomposite materials has the potential to reduce the negative impacts on the environment caused by synthetic packaging [11].

Proteins, polysaccharides and their derivatives are the most extensively utilized biodegradable polymers for packaging due to their abundant availability and ability to polymerize into brittle and hard materials. Polysaccharides and proteins films are most commonly manufactured by a casting process, in addition to extrusion and molding as other approaches. Biodegradable synthetic polymers possess a variety of physical, chemical, and mechanical properties that make them ideal for packaging. These synthetic materials are normally biodegraded through slow chemical hydrolysis in an aqueous environment which can be aided by enzymatic catalysis [12]. Polyhydroxyesters, such as polyglycolic acid; polylactic acid (PLA); and its co-polymers polylactideglycolide have been widely used in antimicrobial packaging. Plasticizers including glycerol, polyethylene glycol, sorbitol, propylene glycol, ethylene glycol, and others are blended with biodegradable polymers to improve the flexibility, extensibility, and moisture sensitivity of packaging films. Biodegradable films incorporating nanoparticles have greater potential for active packaging with an extended shelf life and effective storage of packaged food [4].

#### 1.1.3. Paper

Paper is a commonly used material in food labeling. Paper is the most suitable choice for packaging materials due to its low cost and weight, wide availability, printability, and good mechanical qualities. The main disadvantage is that it is susceptible to humidity and moisture absorption. Various additives are added to paper, paperboard and recycled paper during their manufacturing process that may interact with food ingredients and cause detrimental health effects. The barrier qualities of paper are enhanced by coating its surface with a biopolymer, thereby enhancing its hydrophobicity and usefulness, thus reducing the use of toxic additives [13].

### 1.2. Foodborne Pathogen

Foodborne pathogens (bacteria, parasites, etc.) are biological agents that cause food poisoning [14]. Pathogens that multiply in the host after the consumption of food—and the toxins they generate in food products that are consumed by the host—cause foodborne disease. Accordingly, foodborne disease is divided into two categories: infection and intoxication. Foodborne infection is associated with a prolonged incubation period; therefore, the onset of symptoms in foodborne intoxication is shorter than that in foodborne infection [15]. Most common foodborne disease-causing pathogens, as well as their effects and treatments are given in the table below (Table 1).

### 1.3. Antimicrobial Agents-Overview

One of the most prevalent strategies for minimizing the risk of foodborne illnesses is to develop food packaging materials that are incorporated with antimicrobial agents to improve food quality and safety. Natural antimicrobial agents have to be incorporated in considerable amounts to compensate for the loss of antimicrobial action over time, which increases cost and damage to the sensory properties of foods.

Synthetic antimicrobial peptides (AMP) are not only effective against a wide range of bacteria, but they also have the advantages of low cost, high effectiveness, and biodegradability. Synthetic antimicrobial peptides have more stable characteristics than natural antimicrobial peptides. The copolymerization of synthetic antimicrobial peptide is carried out by ring-opening polymerization of amino acid N-carboxyanhydrides directly with polymers in order to form a packaging film [22]. Synthetic organic antimicrobial agents that are generally used in the food industry are chelating agents, antifungal agents, and preservatives. EDTA exerts its antimicrobial action through the disruption of bacterial cell membranes through the complexion of divalent cations, which function as salt bridges between membrane macromolecules, such as lipopolysaccharides. Imazalil is widely used in food packaging to control a variety of fungal contamination. Preservatives such as propyl paraben, butylated hydroxytoluene (BHT), butylated hydroxyanisole (BHA), etc. can inhibit the growth of molds and yeasts. The mechanism of action these preservatives includes interference with cellular membrane transfer processes and the inhibition of DNA, RNA, and enzyme synthesis in bacterial cells [23].

Overuse of synthetic preservatives results in multidrug-resistant bacteria. Although bacterial growth is more efficiently inhibited by synthetic antibacterial packaging, their toxicity should not be overlooked. Several of synthetic preservatives, such as nitrates, benzoates, sulfites, sorbates, parabens, formaldehyde, BHA, BHT, and others have potentially fatal adverse effects such as hypersensitivity, allergy, asthma, hyperactivity, neurological damage, and cancer [24].

Natural antimicrobial compounds are of greater value, as the green additives in nutritious and functional food formulations and their packaging are in high demand among consumers. Antimicrobial peptides, enzymes, essential oils (EOs), organic acids, and biopolymers such as chitosan are important natural antibacterial agents in the food sector [25].

Essential oils have been explored extensively as potential sources of natural antimicrobial agents in food packaging due to their antifungal, antiparasitic, antibacterial, and antiviral properties. The antibacterial mechanism of action of EOs depends on the type of EO and the microorganism strain. Gram-negative bacteria have a thick lipopolysaccharide membrane that decreases microorganism susceptibility to EOs, but Gram-positive bacteria lack a lipoteichoic acid barrier. As a result, EOs more easily enter into Gram-positive bacteria than Gram-negative bacteria. Various studies have shown that the bioactive components in EOs adhere to the cell surface and enter the phospholipid bilayer of the cell membrane, resulting in membrane damage and detrimental effects on cell metabolic functions, and consequent cell death. The loss of essential intracellular components such as proteins, reducing sugars, ATP, DNA, and the blocking of ATP synthesis and associated enzymes, results in electrolyte leakage and cell death [26].

In the food sector, antimicrobial peptides (AMPs) are a form of natural antibacterial agent. In recent years, biologically derived antimicrobials have sparked interest, particularly for their antilisterial properties. Bacteriocins are peptidic antimicrobial chemicals that are produced by bacteria that have a bactericidal effect against other species of bacteria. Bacteriocins are produced mostly by lactic acid-generating bacteria, making their use in the restriction of specific bacterial growths in food. Quality, safety, and shelf life of food products can be improved by using antimicrobial films containing bacteriocins. Bacteriocins can be divided into four groups: class I, II, III and IV including nisin (lantibiotics);pediocin (non-lantibiotics);lysostaphin (heat-sensitive); and plantaricin S, respectively. Similar to EOs, AMPs target the external and internal processes of bacteria, resulting in membrane damage and cell disruption [27].

Conventional food additives include organic acids that are synthesized by plant and animal metabolism as natural antibacterial agents. Organic acids, such as propionic acid, lactic acid, malic acid, sorbic acid, and tartaric acid are involved in the reduction in pH and disruption of key processes of food products responsible for their antimicrobial action. Organic acids are safe to use in food compositions since they have no detrimental influence on the sensory qualities of food [28].

Chitosan is the one of the most commonly used biocompatible and environment friendly biopolymer possessing inherent antibacterial qualities. Chitosan is formed from chitin (poly-(-14)-2-amino-2-deoxy-D-glucopyranose), which is derived from the deacetylation of chitin that is obtained from the exoskeletons of insects, lobsters, shrimp, and crabs [29]. The source, structure, molecular weight, physicochemical properties, and degree of acetylation have a great impact on the antimicrobial activity of chitosan. Chitosan’s mechanism of action against microorganisms can be characterized as extracellular, intracellular, or both, depending on the targeted site. High-molecular weight (MW) chitosan functions as a chelator of essential metals, preventing nutrients from being taken up extracellularly, and affecting the cell permeability of microorganisms [30]. High-MW chitosan exhibits mean minimum inhibitory concentration (MIC) values in the range of 0.010% to 0.015%*w/v*, respectively. Low-MW chitosan, on the other hand, has both extracellular and intracellular antibacterial activity, affecting RNA, protein synthesis, and mitochondrial function. Chitosan is a strong candidate for antibacterial packaging film due to its film-forming ability, antibacterial activity, and biodegradability [31].

## 2. Types of Food Packaging

### 2.1. Active Packaging

The term “active” in active packaging refers to packaging with superior functionalities, such as managing the atmosphere and preventing microbial growth, which is achieved using innovative engineering materials. AP increases the shelf life and safety of foods by killing foodborne pathogens that promote spoilage. Petroleum-based polymers are commonly utilized in the production of AP [32]. Thermoplastic polymers are superior among petroleum-based polymers due to their unique features, such as fewer amorphous and crystalline structures strongly affecting the release of chemicals from polymers into foods, which is a crucial factor in AP design. Despite all the benefits of petroleum-based polymers (synthetic), such as their capacity for industrial production, low cost, and ability to be converted into various forms, there are a few issues with their use in the food industry, such as their chemical nature and the resulting environmental issues. Recently, there has been a surge in interest in employing biopolymer-based materials to develop APs, as these materials are easy to produce, edible, environmentally friendly, and made from renewable resources. The weak mechanical properties of biopolymer-based packaging are their major drawback, but this can be addressed by using nanomaterials to create effective packaging barriers [3].

### 2.2. Smart Packaging

Smart packaging is designed to track the product, sense the internal or external environment of the package, and communicate with the consumer, as opposed to “active” packaging, which is meant to improve food safety and quality and extend shelf life. Therefore, smart packaging is one that monitors the quality or safety state of the food product and can give a consumer or food manufacturer an early warning. Smart devices, which are small, low-cost tags or labels that are capable of obtaining, storing, and transmitting information on the functions and attributes of packaged food are part of an intelligent packaging system. Smart packaging systems deliver information to the consumer about the condition of food or its environment (temperature, pH). It is an extension of the communication role of conventional packaging, and it communicates with the consumer through its ability to identify, interpret, and record changes in the product’s environment. Hazard Analysis and Critical Control Points (HACCP) and Quality Analysis and Critical Control Points (QACCP) systems, which are developed to detect unsafe food on-site, identify possible health hazards, and develop techniques to minimize or eliminate their occurrence can benefit from smart packaging. Smart packaging also aids in identification of processes having a significant impact on food quality [33]. The attractive benefits associated with smart packaging systems in terms of superior safety, logistics, and marketing indicate that this can emerge to be the most widely used packaging for food in the coming years. Furthermore, there is still a disconnect between research and laboratory solutions, and real-world goods. In this industry, further customization of the packaging system is required to achieve optimal activity or capacity of the various smart packaging requirements [34].

## 3. Method of Preparation of Packaging Films

### 3.1. Packaging Film Formation by Casting Method

The casting method, also known as solvent casting, is the most widely used approach for film formation at laboratory and pilot scales. The process of biopolymer film preparation involves solubilization of the biopolymer in a suitable solvent and casting of the solution in the mold, followed by drying of the casted solution (Figure 1).

The selection of the polymer or polymeric mixture that will comprise the basic film is the first step. The chosen polymer is dissolved in a suitable solvent, and this is important as the film formation ability mainly depends on the polymer’s solubility rather than its melting properties. The resulting solution is poured into a predetermined mold or Teflon-coated petri dish during the casting process. The drying process allows the solvent to evaporate, resulting in a polymer layer that adheres to the mold. Air dryers, such as hot air ovens, tray dryers, microwaves, and vacuum driers are utilized for efficient removal of solvents used and effective peeling of film formed. The air-drying technique for casting edible film is critical for enhancing the intramolecular interaction between the polymer chains and achieving an appropriate microstructure [35,36].

### 3.2. Packaging Film Formation by Extrusion Method

The extrusion method is one of the most widely used polymer processing approaches for the generation of polymeric films. This process alters the structure of the materials and improves the extruded material’s physiochemical qualities. In general, the extrusion process can be divided into three steps: (i) feeding, (ii) kneading, (iii) heating at the exit from the machine (Figure 2).

In the first step, the film forming mixture is fed into the feeding zone and compressed with air. This process is often known as a dry process as it uses minimal water or solvents. Plasticizers such as polyethylene glycol or sorbitol are used, ranging from 10% to 60% by weight to increase the film’s flexibility. In the kneading zone, the strain, temperature, and density of the mixture increase. Finally, in the heating process the thermal energy ranges between 120 and 170 °C. This process is dependent on the thermoplastic characteristic of polymers when plasticization and heating occur above the glass transition temperature and minimal water level conditions [36,37].

### 3.3. Packaging Film Formation by Electrospinning Method

The electrospinning method is used to fabricate a nonwoven web of micro- or nanofibers. This approach delivers high-voltage electricity to the liquid solution and a collector, causing the solution to extrude from a nozzle and generate a jet (Figure 3).

In the drying process, the fibers that are produced through the jet are deposited onto the collector. Electrospinning is a quick and easy way to make micro- or nanopolymer fibers. The inclusion of polymers in the electrospinning solution alters its viscosity, conductivity, surface tension, molecular weight, solvent, concentration, and other properties, all of which are important for the electrospinning process. In the electrospinning process, the dispersed fibers self-assemble under the influence of electric charges, which in turn are dependent on mechanical forces and geometric circumstances. Electrospinning was utilized to create nanofiber polymers, such as cellulose, chitosan, alginate, collagen, polyesters, and polyurethanes [38]. The concentration ofthe polymer in the electrospinning solution is in turn influenced by its ionic properties, as they have a significant impact on the fiber’s shape, diameter and homogeneity. Various processing and environmental aspects such as feed rate, field strength, tip-to-collector distance, needle shape and geometry, temperature, humidity, and airflow have to be considered during optimization of the electrospinning process [39].

## 4. Antimicrobial Packaging System

Antimicrobial packaging systems are developed by loading or coating antimicrobial substance onto polymeric packaging films. Depending on the antimicrobial agent used and its interactions with the packaging material and food composition, the antimicrobial packaging solution can be divided into two categories: (1) those containing an antimicrobial component that migrates to the food’s surface (migrating film); (2) those that are effective against surface microbial growth without migration (non-migrating film) [40].

Antimicrobial packaging extends the shelf life, safety, and quality of many foods by decelerating the migration of antimicrobial agents from a high concentration area (packaging material) to a low concentration area (food). This reduces microbial growth in non-sterile foods and reduces the post-contamination risk of pasteurized products (food). Antimicrobial packaging is designed to complement the existing safety and quality control of the food industry [41].

Preservatives that are incorporated into antimicrobial films provide advantages over those that are added to the food in the sense that preservatives that are present in packaging material exhibit little contact with the food. The slow migration of antimicrobial agents from the packaging material to the product surface could enable their more efficient use in a high concentration if needed. Another benefit of antimicrobial packaging is the lack of loss due to leaching into the food matrix and cross-reaction with other food components, such as lipids and proteins. As a result, the controlled release of antimicrobials into the food allows not only the immediate suppression of undesired microbes, but also long-term residual activity during food transport and storage [42].

## 5. Nanomaterials in Food Packaging

Food packaging research and development has shown tremendous interest in nanomaterials in the last decade due to their non-toxicity, higher thermal stability, and ability to incorporate vital nutrients. Many nanoscale materials are employed to increase the reinforcement of gas/moisture barriers in food packaging, promote the processability of polymers, and to confer specialized capabilities of antimicrobial activity and gas sensing feature [43]. Nano-based “smart” and “active” food packaging provides numerous benefits ranging from superior packaging material with improved mechanical strength, barrier properties, and antimicrobial films, to nano-sensing for pathogen detection and alerting consumers to the safety status of food, over conventional packaging methods [44].

Nanoparticles, nanofibers, and nanocomposites are the three types of nanomaterials that are used in food packaging. Metal nanoparticles, metal oxide nanoparticles, mixed metal oxide nanoparticles, nanoclay families, and carbon materials [carbon nanotubes, graphene] are some of the most prevalent nanomaterials currently under investigation. Nanostructured materials in food packaging can increase the mechanical, chemical, structural, and barrier properties (O_2_/H_2_O, microbial, bacterial, etc.) of the resultant films [45].

Materials with a larger surface-to-volume ratio in the nanoscale range can adhere more copies of the microbe, resulting in improved efficiency. Antimicrobial nanomaterials are particularly intriguing due to their barrier qualities and desired structural integrity, which reduce spoiling and pathogenic microbe development. Nanomaterials can be used as growth inhibitors, killing agents, or even antibiotic transporters in antimicrobial films. Each nanomaterial has diverse applications in food packaging due to basic differences in their structure and physicochemical features. The practical applications of different nanomaterials in the food packaging sector have been the subject of numerous enlightening studies. Despite the antibacterial properties of packaging, it is crucial to note the nanomaterial’s interactions with food ingredients and the potential for food quality changes ranging from sensory characteristics to safety concerns. This is mostly determined by the nanomaterial concentrations that are employed in package production. Antibacterial capabilities can be preserved without compromising food quality or safety on the basis of optimum dose and concentration for controlled nanoparticle release [46].

Several studies on the physical, microbiological, and chemical effects of nano-food packaging have been reported in recent years, owing to their growing importance. This review attempts to provide an overview of the applications of nanoparticles in food packaging and their toxicity.

## 6. Antimicrobial Nanocomposites Nanomaterials Used for Active Packaging

In the food packaging industry, nanocomposites that combine different food packaging materials with nanoparticles are growing in popularity. Apart from their exceptional antibacterial spectrum, they possess good mechanical performance and robust resistance qualities [47]. They are multiphase materials formed by combining a matrix (continuous phase) with a nano-dimensional material (discontinuous phase). The nano-dimensional phase is classified as nanospheres, nanowhiskers or nanorods, nanotubes, nanocrystalsand nanosheets, depending on the nanomaterial [48]. Nanocomposite materials serve as innovative, high-performance, lightweight, and environmentally friendly composite materials in place of non-biodegradable plastic packaging materials. Biopolymers such as chitosan, carboxymethyl cellulose (CMC), and starch, due to their biodegradability and non-toxicity, could be used to mitigate environmental concerns [44]. Various nanomaterials such as silica, clay, organo-clay, graphene, polysaccharide nanocrystals, carbon nanotubes, chitosan, cellulose-based, ZnO, CuO, and TiO_2_ nanoparticles have been investigated as fillers [49].

### 6.1. Nanoparticles

Nanoparticles are particles with one or more dimensions in the size range of 1 to 100 nm that have superior qualities, such as high reactivity, sensitivity, surface area, stability, and strength over larger materials [50]. It has been demonstrated that the antibacterial activity of the inorganic nanoparticles can cover a wide range of microorganisms, including that of food-borne diseases. The antimicrobial agents, such as titanium oxide, silver, zinc oxide, copper, and gold are associated with high efficacy at low concentrations and can therefore replace traditional chemical antibacterial materials [51]. In recent years, studies have focused on organic nanoparticles such as chitosan, zein, and nanocellulosic nanoparticles for their mechanical and barrier capabilities when used as packaging materials [52].

#### 6.1.1. Halloysite Nanotubes

Jang et al., fabricated packaging paper containing thyme essential oil (TO)-loaded halloysite nanotubes (HNTs) and nanocapsules with inner diameters of 15 nm and outer diameters of 50 nm by vacuum pulling method followed by end-capping or a layer-by-layer surface coating strategy for complete encapsulation. Surface coating of HNTs exhibited considerably sustained and long-lasting release qualities at room temperature, compared to untreated and end-capped HNTs. The antimicrobial activity of the packaging paper containing TO-loaded HNT nanocapsules were tested for 25 days against *E. coli*; it was found to exhibit good activity for the first 10 days with the bacterial count reduced to ~2.5 log CFU/cm^2^, while the untreated paper packaging showed *E. coli* growth of 10^5^ CFU/cm^2^. The packaging paper proved to be particularly effective in eradicating *E. coli* within the initial 5 days with the bacterial count reduced to ~1.5 log CFU/cm^2^ [53].

#### 6.1.2. Gold Nanoparticles

Chowdhury et al., fabricated poly(vinyl) alcohol (PVA) crosslinked composite films incorporated with gold nanoparticles (AuNPs) and graphene oxide (GO), along with glyoxal and/or glutaraldehyde (GA) as a crosslinking agent by the casting method. Various films were prepared using PVA, PVA-Glyoxal, PVA-Glyoxal-AuNPs, PVA-GA and PVA-GA-GO, and exhibited a tensile strength of 0.47 ± 0.41 Mpa; 1.07 ± 0.05 Mpa; 1.45 ± 0.07 Mpa; 0.66 ± 0.03 Mpa; and 1.51 ± 0.07 Mpa, respectively. Young’s modulus exhibited values of 0.41 ± 0.02 Mpa; 1.12 ± 0.05 Mpa; 1.45 ± 0.07 Mpa; 0.66 ± 0.03 Mpa; and 1.55 ± 0.08 Mpa, respectively. The water vapor transmission rate was 54.24 ± 2.67 g/m^2^ h; 36.68 ± 2.24 g/m^2^ h; 33.11 ± 1.65 g/m^2^ h; 38.49 ± 2.11 g/m^2^ h; and 32.13 ± 1.73 g/m^2^ h, respectively. The water solubility was 41.86 ± 3.56%; 35.67 ± 4.36%; 45.35 ± 4.70%; and 47.65 ± 4.24%, respectively. The antimicrobial activity of the PVA-glyoxal and PVA-GA films was increased on addition of AuNPs that were added to the resulting 13 mm inhibition zone against *E. coli*, whereas the PVA-GA-GO film was demonstrated by the formation of a 10 mm inhibition zone against *E. coli*. The antimicrobial activity of the film was attributed to GO and AuNPs. However, AuNPs possess a greater antimicrobial capability than GO [54].

#### 6.1.3. Graphite Carbon Nitride Nanosheets/Molybdenum Sulfide Nanodots

Ni et al., developed antibacterial films that were composed of graphite carbon nitride nanosheets/Molybdenum sulfide nanodots (CNMo) were loaded into konjac glucomannan (KG) films by the casting method. The film containing 10% CNMo (KCNMo-10) exhibited the highest tensile strength and thermal stability. The film exhibited antibacterial efficacy against *S. aureus* and *E. coli* with an inhibition zone measuring ~2.1 cm and ~1.3 cm, respectively. Cherry tomatoes that were packed with KCNMo-10 films remained intact at the end of 18 days study, as compared to the unpacked group and PE group. Furthermore, in vitro cells and hemolysis investigation revealed that the hemolysis ratios of fresh and film-coated cherry tomatoes, relatively, were both less than 5%, while the cell activity of the film-coated group remained greater than 95%, indicating that the film was safe [55].

Ni et al., developed chitosan/negatively charged graphitic carbon nitride-self-activation bionanocomposite films using one-step electrostatic self-assembly. The film after self -activation under visible light exhibited an inhibitory rate of 99.8 ± 0.26% and 99.9 ± 0.04% against *E. coli* and *S. aureus*, respectively. In comparison to neat chitosan films and commercially used PE films, this bionanocomposite film effectively extends the shelf life of tangerines to 24 days. The film was proven to be safe and innocuous by hemolysis and cytotoxicity experimentation [56].

#### 6.1.4. Magnesium Oxide Nanoparticles

Wang et al., fabricated a carboxymethyl chitosan (CMCS) and nano MgO nanocomposite film for packaging of water-rich food product. CMCS/MgO composites have greater thermal stability, UV shielding performance, and water insolubility than pure CMCS. The increase in MgO content in the film caused improvement in their physical attributes such as elasticity and ductility of CMCS at low filler concentrations (1.0 wt.%). CMCS/MgO composites had outstanding antibacterial action against *L. monocytogenes* and *Shewanella baltica* in terms of biological characteristics. The increase in MgO content upto 1% in the composite film caused 99.99% inhibition rate against *L. monocytogenes* and *S. baltic* [57].

#### 6.1.5. Palladium and Platinum Nanoparticles

Chlumsky et al., developed palladium (PdNPs) and platinum (PtNPs) nanoparticles by cathodic sputtering. The antimicrobial activity of both nanoparticles exhibited reduced viable cells by 0.3–2.4 log CFU/mL (PdNPs) and 0.8–2.0 log CFU/mL (PtNPs), respectively, with average inhibitory rates of 55.2–99% for PdNPs and 83.8–99% for PtNPs against *E. coli*, *S. enteric Infantis*, *L. monocytogenes*, and *S. aureus*. PdNP concentrations of 22.25–44.5 mg/L and PtNP concentrations of 50.5–101 mg/L were found to be the most effective in reducing biofilm formation. The IC_50_ values of PdNPs (>4.45 mg/L) and PtNPs (>10.1 mg/L) were determined in human epithelial kidney cells, human dermal fibroblasts, human keratinocytes, primary human coronary artery endothelial cells and primary human renal tubular epithelial cells using in vitro cytotoxicity studies [58].

#### 6.1.6. Polymeric Nanoparticles

Wu et al., developed chitosan/ε-polylysine (ε-PL) bionanocomposite films with sodium tripolyphosphate (TPP) as a cross-linking agent using an in situ self- assembly technique. The four films were prepared by varying the concentration of the ingredients that exhibited physical properties such as thickness, water solubility, water vapor permeability, tensile strength, and elongation at breaking in the range from 0.082 ± 0.003 to 0.090 ± 0.007 mm; 18.23 ± 0.21 to 21.64 ± 0.53%; 2.42 ± 0.14 to 3.13 ± 0.14 ×10^−10^ g/(s·m·Pa); 17.09 ± 0.58 to 23.30 ± 0.22 MPa; and 32.18 ± 0.78 to 44.63 ± 2.60%, respectively. The bionanocomposite films displayed good antibacterial action against *E. coli* and *S. aureus* with an inhibition zone ranging from ~20 to ~8 mm and ~13 to ~7 mm, respectively. The realese of ε-PL films followed sustained pattern and was proportional to the TPP concentration. As the concentration of ε-PL increased proportionally, increase in antibacterial activity was observed suggesting synergism in antimicrobial activity of chitosan and ε-PL in the film [59].

Roy et al., developed curcumin incorporated polylactic acid films using a solution casting method. On addition of curcumin, the mechanical properties, such as thickness (54.9 ± 3.5 to 64.6 ± 1.8 µm); tensile strength (40.8 ± 1.6 to 49.4 ± 1.0 MPa); elongation at break (5.7 ± 1.1 to 7.9 ± 1.8%); and Young’s modulus (0.73 ± 0.1 to 1.00 ± 0.1 GPa) were enhanced slightly without changing the thermal stability of the PLA film. The water vapor permeability (2.02 ± 0.13 to 2.15 ± 0.44 ×10^−11^ g·m/m^2^·Pa·s) and water contact angle (71.6 ± 5.0 to 74.3 ± 2.9°) of the PLA film were slightly increased by the addition of curcumin. The PLA/curcumin composite film (50 μg/mL curcumin) demonstrated antioxidant activity of 54.1 ± 1.4% and 66.4 ± 1.6%, respectively in 2,2-diphenyl-1-picrylhydrazyl (DPPH) and ABTS tests. Moreover, films exhibited a distinctively slower growth of both *E. coli* and *L. monocytogenes* in terms of 1–2 log cycles lower viable colony forming units than the control group. The antimicrobial activity of the film was due to curcumin incorporated in the films [60].

Cesur et al., developed biodegradable polycaprolactone (PCL) food packaging films that were incorporated with organo-nano clay (C) (0.4 wt.%); chitosan (25, 50, and 75 wt.%); and glycerol mono-oleate (G) or oleic acid (OA) as a plasticizer (5, 10, 20, and 30 wt.%). The mechanical properties of the film such as Young’s modulus, tensile strength and elongation at break were found to be in the range of 0.10 ± 0.03 to 3.49 ± 2.99 MPa; 0.16 ± 0.12 to 1.13 ± 0.01 MPa; and 1.5 ± 0.4 to 486.9 ± 40.5%. Films of varying concentration of plasticizer and chitosan exhibited antimicrobial properties against *E. coli*, *P. aeruginosa* and *C. albicans*. These microorganisms were found to be resistant to the composite film while *B. cereus* was unaffected by any of the films [61].

#### 6.1.7. Silica/Nanoclay/Montmorillonite Nanparticles

Zhang et al., developed a cinnamon essential oil (CEO)-loaded mesoporous silica nanoparticles (MSNPs) that were incorporated into potato starch films by a casting method and exhibited good physical and mechanical properties, including water vapor transmission rate (644.41 g d^−1^·m^−2^); oxygen transmission rate (4.86 g d^−1^·m^−2^); tensile strength (56.12 ± 1.39 MPa); elongation at break (50.00 ± 1.25%); and thickness (25.93 ± 0.83 µm). The minimum inhibitory concentration of MSNP-CEO used in the film was 6 mg/mL. The developed films exhibited significant antimicrobial activity against the *Mucor* species than against the *Mucor circinelloide s* strain which was attributed to the CEO that is encapsulated in MSNPs [62].

Ellahi et al. developed a polypropylene film that was coated with silica nanoparticles and *Pistacia atlantica* tree gum essential oil (GEO). The antibacterial activity of the produced packing film containing 0.001 g of silica nanoparticle encapsulating GEO had a greater inhibitory impact (3.45 log CFU/g) against *S. aureus*, *S. enterica*, *E. coli*, and *L. monocytogenes*, while a package without silica nanoparticles exerted no antibacterial activity. The shelf life of milk was extended by 35 days due to the extended release of GEO from nanoparticles [63].

Wu et al., fabricated curcumin-loaded mesoporous silica nanoparticles(CMSNP) that were integrated into chitosan films using the solvent casting process. The thickness, tensile strength, elongation at break and water vapor permeability of the film were found to be 0.0931 ± 0.0021 mm; 19.87 ± 1.02 MPa; 25.46 ± 2.16%; and 15.21 ± 1.83 g 10^−11^/s m^2^ Pa, respectively. The CMSNP film and plain Chitosan/Curcumin blend film exhibited zone of inhibitions (ZOI) of ~7.5 mm; ~8 mm and ~8 mm; ~10 mm against *E. coli* and *S. aureus* respectively. Higher antimicrobial activity of MSNP film was attributed to slow controlled release of curcumin from the film [64].

Jha developed corn starch-chitosan nanoclay bionanocomposite films that were incorporated with glycerol/sorbitol as a plasticizer and potassium sorbate/grapefruit seed extract as an antimicrobial agent. The mechanical properties of the films such as tensile strength, elongation at break and water vapor permeability ranged from 12.7 to 19.2 MPa; 56.15 to 77.88%; and 3.3 × 10^−11^ to 9.5 × 10^−11^ g/m·s·Pa. The bionanocomposite films exerted antifungal activity against *Aspergillus niger* with the inhibition zone ranging from 13.47 ± 0.79 mm to 47.10 ± 0.50 mm and also exhibited good antifungal protection to the bread samples for 20 days [65].

Benhacine et al., developed poly(e-caprolactone)/silver exchanged montmorillonite nanocomposite films using solvent casting method. The quantity of silver ions released from nanocomposite estimated by atomic absorption spectroscopy was found to increase over time after 30 days of immersion in slightly acidified water. Furthermore, due to the presence of long-lasting biocidal silver nanoparticles, the developed films exhibited significant antibacterial efficacy against *S. aureus*, *E. coli*, *Salmonella*, and *P. aeruginosa* and exhibited bacterial growth inhibition greater than 24 h [66].

Dairi et al., developed nanobiocomposite films containing silver nanoparticles (AgNPs)/gelatin-modified montmorillonite nanofiller, thymoland plasticized cellulose acetate/triethyl citrate. The nanobiocomposite films exhibited elastic modulus, tensile strength, elongation at break, and oxygen transmission rate in the range of 1625 ± 144 to 2436 ± 74 MPa; 36.64 ± 1.94 to 48.28 ± 2.42 MPa; 2.55 ± 0.24 to 7.77 ± 1.86%; and 48.10 ± 3.67 to 66.82 ± 2.54 cm^3^·mm·m^−2^·day^−1^ respectively. The films demonstrated antimicrobial activities against *E. coli* (ZOI = 28 ± 0.8 mm); *S. aureus* (ZOI = 25 ± 0.5 mm); *Salmonella* sp. (ZOI = 20 ± 0.2 mm); *P. aeruginosa* (ZOI = 25.5 ± 1 mm); *A. niger* (ZOI = 22.5 ± 0.6 mm); and *A. flavus* (ZOI = 19 ± 0.2 mm), with *E. coli* being the most sensitive. The antimicrobial activity of the film was attributed to the thymol and AgNPs loaded into monmorillonitenanofiller. Moreover, thymol exhibited additional antioxidant activity of with percent scavenging in the range of 86.38% to 90.55% [67].

Zhu et al. fabricated polylactic acid filmscontaining mesoporous silica nanoparticles-loaded with clove essential oil by the solvent volatilization method. The result of antimicrobial evaluation on *Agaricusbisporus* (white button mushrooms) that were packed in developed antimicrobial film, revealed fewer colony-forming units of various bacteria such as mesophilic bacteria, psychrophilic bacteria and *Pseudomonas sps*, as compared to that of plain PLA films. The presence of clove essential oil in the film effectively suppressed the growth of microbes and extended the shelf life of white button mushrooms [68].

Mondal et al., fabricated polybutylene adipate-co-terephthalate (PBAT)/cetyltrimethylammonium- modified montmorillonite (CMMT)- based nanocomposite film that was doped with sodium benzoate (SB) as an antibacterial agent by solution mixing method. Compared to the PBAT film, the PBAT/CMMT nanocomposite film exhibited superior barrier capabilities against water vapour (4.5 × 10^−5^ g/cm^2^/d) and methanol vapour (3.94 × 10^−4^ g/cm^2^/d). The nanocomposite film inhibited *B. subtilis* and *S. aureus* with inhibition zone measuring 20 and 21 mm, respectively. The antimicrobial activity of PBAT/CMMT/SB was found to be superior to the PBAT/SB film [69].

#### 6.1.8. Silver Nanoparticles

Brito et al., developed silver nanoparticles (1.5, 3.75, 7.5, 15, 30, 60, and 75 mg/mL) that were loaded on low-density polyethylene films by extrusion method. The nanostructured films exhibited antimicrobial properties against *S. aureus*, *E. faecalis*, *E. coli*, *S. typhimurium*, and *P. expansuma* with a reduction in colony forming units from 2.035 to 1.682 log CFU/mL; 1.493 to 0.934 log CFU/mL; 2.072 to 0.279 log CFU/mL; 1.625 to <1 log CFU/mL; and <1 log CFU/mL, respectively with the increase in concentration of silver nanoparticles [70].

Tripathi et al., synthesized a biodegradable poly (vinyl alcohol)-biogenic silver antibacterial nanocomposite film. Silver nanoparticles were synthesized using *Ficusbenghalensis* leaf extract, which is a cost-effective, environmentally friendly, and quick approach. The antibacterial effectiveness of the PVA-biogenic silver nanocomposite film against *Salmonella typhimurium* was excellent, with inhibition zones ranging in diameter from 9 to 18.3 mm. Silver nanoparticles were suggested to be responsible for the antimicrobial activity of the film [71].

#### 6.1.9. Titanium Dioxide Nanoparticles

Siripatrawan et al., developed active packaging from chitosan and TiO_2_ NPs in a series of concentrations (0, 0.25, 0.5, 1, and 2% *w/w*). The chitosan film containing 1% TiO_2_ NPs demonstrated antibacterial action against Gram-positive (*S. aureus*) and Gram-negative (*E. coli*, *S. typhimurium*, and *P. aeruginosa*) bacteria and fungi (*Aspergillus* and *Penicillium*). Thus the result suggested that chitosan-TNPs nanocomposite films could be used as an active packaging [72].

#### 6.1.10. Zinc Oxide Nanoparticles

Hu et al., developed composite film containing ZnO-chitosan nanoparticles of an average size of 25 nm that were incorporated into modified starch matrix using sol-gel synthesis. The films that were loaded with nanoparticles exhibited a significant reduction in water vapor permeability from 51.0 to 43.7%, and an increase in tensile strength from 4.11 to 12.79 MPa. The films exhibited stronger antimicrobial activity against *S. aureus* (percent inhibition of 100%) than *E. coli* (percent inhibition of 65%) [73].

Liu et al., developed chitosan biopolymeric film loaded with antioxidant extract of bamboo leaves (AOB) and zinc oxide nanoparticles (ZnO) by the casting method for active food packaging. The thickness, tensile strength, moisture content and water vapor permeability of the film was found to be in the range of 0.032 to 0.088 mm; 12.43 MPa; 11.95 ± 0.80 to 28.89 ± 5.24%; 1.20 ± 0.07 to 2.21 ± 0.01 g∙mm/m^2^ day∙kPa, respectively. The developed films exhibited antibacterial activity against *S. aureus* and *E. coli* with zone of inhibition ranging from 27.01 ± 1.28 to 28.54 ± 3.55 mm and 26.60 ± 3.00 to 29.69 ± 2.53 mm, respectively as compared to CS/ZnO film with zone of inhibition 22.85 ± 6.81 mm and 23.51 ± 3.12 mm against *S. aureus* and *E. coli*, respectively. The antimicrobial activity of the film was attributed to synergism between chitosan and ZnO nanoparticles. In addition, the inclusion of AOB considerably reduced UV light transmittance thereby significantly increased the antioxidant activity (87.93%) of the films [74].

Wu et al., developed biocomposite films based on soy protein isolate (SPI) that were incorporated with cinnamaldehyde (CIN) and zinc oxide nanoparticles. The thickness ofthe SPI + CIN + ZnO nanocomposite film was 1.3 times (89.2 ± 3.5 μm) than that of the pure SPI film (70.6 ± 1.5 μm).The neat SPI film sample exhibited tensile strength of 1.68 MPa and breaking elongation of 133.6%, whereas the SPI + CIN + ZnO film showed tensile strength of 2.11 MPa and breaking elongation of 164.0%, which were 1.26-fold and 1.23-fold higher than the pure SPI film, respectively. SPI + CIN + ZnO film exhibited 66.1% oxygen permeability and 54.8% water permeability, compared to plain SPI film. The SPI + CIN + ZnO films showed 1.56-fold and 1.24-fold greater antifungal activity than the SPI + ZnO and SPI + CIN films, respectively [75].

Janani et al., fabricated bioactive nanocomposite films that were composed of tragacanth; polyvinyl alcohol; ZnO nanoparticles; and ascorbic acid (AA) using glycerol as a plasticizer and citric acid as a cross-linker by casting method for food packaging. The incorporation of AA and ZnO NPs into nanocomposite films improved antioxidant activity from 50% to 66%. The nanocomposite films exhibited higher antibacterial activity against *S. enterica* (ZOI range 13.5 to 18); *L. monocytogenes* (ZOI range 10.5 to 17.25); *Y. enterocolitica* (ZOI range 13.5 to 17.25); *P. aeruginosa* (ZOI range 16.5 to 18.25); *E. coli* (ZOI range 12 to 16.5); and *S. aureus* (ZOI range 15.75 to 18). The antimicrobial activity was attributed to ZnO NPs in the films [76]

Zare et al., fabricated zinc oxide−silver nanocomposites (ZnO-AgNCs) using *Thymus vulgaris* leaf extract as a stabilizer and reducing agent integrated into poly(3-hydroxybutyrate-co-3-hydroxyvalerate)-chitosan (PHBVCS) films by solvent casting method. The nanocomposite biopolymer films exhibited improved mechanical properties such as Young’s modulus (2.5–4.5 GPa); tensile strength (10–35 MPa); and elongation at break (1–7%) as compared to the neat PHBV-CS films. The nanocomposite biopolymer films exhibited antibacterial activity against *E. coli* (ZOI = ~12–22 mm) and *S. aureus* (ZOI = ~14–25 mm). The result of in vitro cytotoxicity study revealed that the hemolytic activity was less than 3% hemolysis even at a higher concentration of the developed ZnO-Ag NCs biopolymer films. The result suggested that ZnO and Ag synergistically improved the antimicrobial activity without any associated toxicity [77].

S.K. et al., fabricated nanocomposite film by combining Mahua oil-polyol based polyurethane (PU), chitosan, and ZnO nanoparticles by solvent casting method. PU/CS/nano ZnO(5%) composite film exhibited thickness of 0.1 mm, water vapor transmission rate of 163 g/m^2^/day, oxygen transmission rate of 1883.15 cm^3^/m^2^/day, tensile strength of 8.1 MPa, and elongation at break of 2.156%. The developed film was more resistant to *E. coli* (25 mm) than to *S. aureus* (20 mm). After 28 days in the soil, the weight loss of the PU/CS/5%nanoZnO biodegradable film was 86%. Carrot pieces that were wrapped with composite film had shelf life extension of up to 9 days. Compared to commercial polyethylene film, the film containing zinc oxide nanoparticles demonstrated significant antimicrobial action [78].

Yadav et al., developed chitosan and zinc oxide nanoparticles loaded gallic-acid films, (CS-ZnO@gal) as environment friendly food packaging material. The thickness of CS-ZnO@gal1; CS-ZnO@gal2; and CS-ZnO@gal3 films were 0.1200 ± 0.0141; 0.1233 ± 0.0093; and 0.1031 ± 0.0276 mm, respectively. CS-ZnO@gal1; CS-ZnO@gal2; and CS-ZnO@gal3 films demonstrated lower water vapor permeability values of 3.065 ± 0.0586; 2.057 ± 0.0657; and 1.176 ± 0.2157 × 10^−6^ g·m^−1^·s^−1^·Pa^−1^, respectively, while the oxygen permeability values were 8.772 ± 0.2091; 8.452 ± 0.3011; and 5.570 ± 0.3051 cc/m·24 h·atm, respectively. Strong antioxidant behavior was exhibited by CS-ZnO@gal1, 2 and 3 films as percent scavenging activity of 64.66%, 66.56% and 68.51%, respectively, tested using DPPH method, and 76.38%, 77.29% and 83.43 by the ABTS method respectively. The antibacterial activity of chitosan films was linearly proportional to the amount of ZnO@gal in the composite films; as a result, the Ch-ZnO@gal3 composite film was the most effective against both *B. subtilis* and *E. coli*. The antimicrobial activity of the film was attributed to the synergistic action of chitosan, ZnO and gallic acid [79].

#### 6.1.11. Zein Colloidal Nanoparticles

Kang Li et al., fabricated thymol and sodium caseinate (SC) (emulsifier and stabilizer)-loaded zein colloidal nanoparticles as antimicrobial films using an antisolvent method. ZP2 (20%); ZP3 (30%); and ZP4 (40%) showed an inhibition zone ranging from 15.89 ± 0.74 mm to 18.81 ± 0.56 mm against *E. coli* and *Salmonella*, whereas ZP0 (0%) and ZP1 (10%) showed no significant action. The results suggested that the amount of thymol in the films had a significant impact on their inhibitory efficacy against *E. coli* and *Salmonella*. The rate of thymol release from nanoparticle-based films followed a two-step biphasic mechanism, with an initial burst effect followed by a slower release, and the inclusion of zein-SC nanoparticles in the film matrices maintained thymol release, indicating that the developed films could be used to increase the shelf life of food products [80].

#### 6.1.12. Nanoemulsions

McDaniel et al., developed antifungal pullulan packaging films containing cinnamaldehyde, eugenol, and thymol as active essential oil compounds (EOC) that were encapsulated in refined coconut oil (liquid) and hydrogenated palm oil (solid) as carrier oils to form liquid lipid nanodroplets and solid lipid nanoparticles, respectively. The antifungal activity of packaging film against *Rhizopusstolonifer, Alternaria* sp., and *Aspergillus niger* revealed that pullulan films containing active nanoemulsions exhibited significant antifungal properties. All the films that were loaded with EOC showed antifungal activity to varying extents, with cinnamaldehyde-loaded solid lipid nanoparticles exhibiting the highest antifungal activity with an inhibition zone measuring 20.3, 15.9, and 18.2 mm against *Alternaria* sp., *A. niger*, and *R. stolonifer*, respectively, while the control films exhibited no antifungal activity [81].

#### 6.1.13. Nanoliposomes

Wu et al., fabricated laurel essential oil (LEO) and AgNPs (Lip-LEO-AgNPs) nanoliposome composite combined with chitosan to form polyethylene (PE-CS) for pork packaging utilizing a film hydration method for controlled release. Antibacterial testing of the film coating against *S. aureus* and *E. coli*, the bacteria that cause decomposition and putrefaction in meat, revealed that PE and PE-CS films showed no antibacterial activity, while PE-CS-Lip/LEO/AgNPs showed inhibition zone of ~5 mm for *S. aureus* and ~1 mm for *E. coli*. The antimicrobial evaluation substantiated the synergy of chitosan, essential oil and silver nanoparticles in th film and thus could be utilized for prolonging the shelf lives of food products [82].

Wu et al., developed gelatin films that were loaded with cinnamon essential oil (CEO) nanoliposomes of 70.53 ± 0.57 µm by thin film ultrasonic dispersion method. The gelatin films that were combined with CEO nanoliposomes exhibited a reduction in tensile strength (6.50 ± 1.07 MPa); water solubility (23.76 ± 1.38%); water content (22.11 ± 1.31%;, and water vapor permeability ((1.96 ± 0.05) × 10^−10^ g/(s·m·Pa)), followed by an increase in elongation at break (85.71 ± 8.16%). The film incorporating CEO nanoliposomes showed an increase in antimicrobial stability against, *E. coli*, *S. aureus*, and *A. niger*, and a decrease in CEO release rate. CEO nanoliposome films resulted in a sustained release effect, demonstrating better control of pathogen compared to the gelatin-CEO film after storage for one month [83].

#### 6.1.14. Nanosponges

Simionato et al., developed cinnamon essential oil that was encapsulated in cyclodextrin nanosponges (α-NS and β-NS) as an antimicrobial active food packaging. Cinnamon essential oil, both alone and in nanosponges, was found to exhibit an antibacterial effect against foodborne pathogens. The MIC values that were obtained ranged from 125 to 500 ppm against *Brochothrix thermosphacta*, *L. monocytogenes*, verotoxigenic *E. coli* and *Y. enterocolitica*, with *B. thermosphacta* being the most susceptible bacteria to CEO. Time-kill experiments revealed that the essential oil, either alone or encapsulated, had a bacteriostatic impact on all the bacteria tested, with the exception of *Yersinia enterocolitica*, which exhibited a bactericidal effect. Furthermore, cinnamon essential oil was efficacious at lower concentrations in culture medium due to the controlled release provided by encapsulation, than when it was simply dissolved in it [84].

Amongst the nanofiller used in antimicrobial active food packaging discussed above, silica nanoparticles/nanoclay/montmorillonite and polymeric nanoparticles are found to be used widely as nanocarriers for antimicrobial agents. Metal/metal oxide nanoparticles such as silver, gold, CuO, ZnO, TiO_2_, etc., are widely used as antimicrobial agents individually or in combination with other antimicrobial agents in food packaging. Among natural antimicrobial agents used for active food packaging, essential oils are most widely used encapsulated in various nanoparticles.

### 6.2. Nanofiber

Nanofibers, produced by electrospinning technology are a popular alternative in the food preservation sector. Electrospinning technology has a number of distinct advantages such as low equipment and experimental costs, high fiber yield, and a large fiber specific surface area (fiber diameters ranging from tens of nanometers to several micrometers), and it can be used to fabricate a wide range of materials. Antibacterial compounds can be encapsulated in electrospun nanofibers and because of their porous structure, they can accomplish a gradual release of antimicrobials with a high carrying capacity [85]. Some of the prominent polymers that are used to prepare nanofibers include cellulose, chitosan, alginate, polyesters, etc. One of the most important areas where electrospun nanofibers can be developed further in the future is food packaging, because antimicrobials, antioxidants, and other bioactive resources can be easily integrated intoelectrospun nanofibers during electrospinning to enhance shelf life and food safety [39].

#### 6.2.1. Cellulose Nanofiber/Nanowhisker/Nanocrystals

Saravanakumar et al. developed an antibacterial polymeric film by combining sodium alginate (3%) as a plasticizer with copper oxide nanoparticles (5 mM)-loaded cellulose nanowhisker (0.5%) using a casting method. The film that was coated on freshly cut pepper demonstrated good antimicrobial activity against *S. aureus*, *Salmonella* sp., *C. albicans*, *E. coli*, and *Trichoderma* sp. with inhibition zone of 27.49 ± 0.91 mm; 25.21 ± 1.05 mm; 23.35 ± 0.45 mm; 12.12 ± 0.58 mm; and 5.31 ± 1.16 mm, respectively. Copper oxide nanoparticles were responsible for exerting antimicrobial action in the film. The film also exhibited significant antioxidant activity in terms of DPPH (46.55%) and ABTS (35.46%) scavenging [86].

Alizadeh-Sani et al., designed cellulose nanofiber- or whey protein matrix-based packaging materials with TiO_2_ nanoparticles and rosemary oil using the casting method and evaluated their antimicrobial activities against resistant foodborne pathogenic microorganisms such as *L. monocytogenes* and *S. aureus* found in meat. The results indicate that the unpacked meat sample exhibited an initial bacterial count of 3.3 log CFU/g, which changed to 7.1 log CFU/g after 9 days of storage at 4 °C. For the same storage time, the PBC values of the packed sample were significantly lower than 4.15 log CFU/g and remained below the microbiologically permissible limit (5.3 log CFU/g) for 15 days. The results indicate that there is a synergistic antimicrobial activity of rosemary oil and TiO_2_ nanoparticles which helps to prolong the shelf life of meat samples [87].

Yang et al., developed a nisin-loaded nanocellulose-based hybrid film from sugarcane bagasse. The different concentration of nisin (640, 1280, 1920, 2560 and 3200 mg/L) influenced the mechanical properties of the film, such as its tensile strength (51.56 ± 5.93, 50.35 ± 6.79, 48.57 ± 2.18, 39.47 ± 6.67 and 28.38 ± 2.55 MPa); Young’s modulus (1.12 ± 0.71, 1.34 ± 0.65, 2.21 ± 0.75, 1.87 ± 0.59 and 1.43 ± 0.2 GPa); and elongation at break (11.22 ± 1.59, 8.06 ± 2.66, 5.98 ± 1.58, 5.29 ± 1.99 and 4.48 ± 1.11%). Cellulose nanofibrils (CNFs)/nisin hybrid films with 1920 mg/L nisin exhibited good mechanical properties and were employed as a liner in low-density polyethylene plastic packaging for ready-to-eat ham, and complete inhibition of *L. monocytogenes* was observed after 7 days at 4 °C storage [88].

Leite et al., fabricated gelatin films incorporating rosin-grafted cellulose nanocrystals (r-CNCs) by solution casting for antimicrobial packaging applications. The gelatin/r-CNCs films had a moderate water vapor permeability of 0.09 g·mm/m^2^·h·kPa, a high tensile strength of 40 MPa, and a high Young’s modulus (1.9 GPa). Antimicrobial nanocellulose was developed by grafting rosin onto CNCs, which prevented the growth of *E. coli* (MIC 22 mg/mL) and *S. aureus* (MIC 5.5 mg/mL). The mozzarella cheese samples that were packed with pure gelatin, r-CNCs gelatin film, and a PVC film were evaluated, and the results demonstrate that microbial deterioration was visible in the control and gelatin-packed cheese samples, particularly in the PVC-packed sample, while the sample that was packed in 6 wt.% r-CNCs-loaded gelatin film had no microbiological growth [89].

Costa et al., fabricated chitosan/cellulose nanocrystals (CNC) films (5, 10, 25, and 50 wt.%) with thickness ranging from 0.090 mm to 0.10 mm by solvent casting as active pads for meat packages to prolong its shelf-life and preserve its properties over time. Chitosan films without CNC had a tensile strength of 7.98 MPa; a 61.4% elongation at break; anda Young’s modulus of 22.4 MPa. On addition of CNC (5, 10, and 50 wt.%), the tensile strength increased gradually to 8.93, 13.0, and 25.3 MPa and the elongation at break was retained, while the Young’s modulus increased to 26.1, 40.2, and 92.7 MPa, respectively. The antimicrobial activity of the film containing pure chitosan, Chitosan + 5 wt.%CNC; Chitosan + 10 wt.%CNC; Chitosan + 25 wt.%CNC; and Chitosan + 50 wt.%CNC against *S. aureus* was 4.57 ± 0.08, 3.7 ± 1.3, Total, Total and 2.9 ± 0.6, respectively; activity against *E. coli* was 3.54 ± 1.3, 1.89 ± 0.9, Total, Total and Total, respectively; and against *C. albicans* was 2.3 ± 0.27, 2.1 ± 0.04, Total, Total and 2.28 ± 0.1, respectively. Finally, chitosan-based films inhibited the growth of *Pseudomonas* and *Enterobacteriaceae* bacteria in meat during the first days of storage when compared to commercial membranes, while chitosan/CNC films inhibited the total volatile basic-nitrogen (TVB-N), implying their effectiveness in preventing meat spoilage under cold storage conditions [90].

Chen et al., developed active films based on pullulan and carboxylated cellulose nanocrystal (C-CNC) that were incorporated with tea polyphenol (TP) by a solution casting method. The film that was exhibited improved the water barrier properties ((1.75 ± 0.04) × 10^−10^ g/(s·m·Pa)); tensile strength (39.07 MPa); and elongation at break (6.23 ± 0.18%) of the resulting bionanocomposite film. Furthermore, the inclusion of TP increased the UV-barrier characteristics, antioxidant activity, and reduced induced transmittance from 73.20% to 0.15%. The film demonstrated enhanced antibacterial activity of PC-TP bionanocomposite films against *E. coli* (12 mm) and *S. aureus* (16 mm). The antimicrobial activity was attributed to the presence of TP in the film [91].

Vilela et al., fabricated an antimicrobial nanocomposite film composed of polysulfobetaine methacrylate (PSBMA) and bacterial nanocellulose (BNC) through one-pot polymerization in the presence of polyethylene glycol diacrylate as a cross-linking agent. The thickness of the films was 131 ± 24 µm for PSBMA/BNC1 and 194 ± 55 µm for PSBMA/BNC2. The films demonstrated good mechanical performance such as the Young’s modulus (4.6 ± 0.5 GPa for PSBMA/BNC1 and 3.1 ± 0.4 GPa for PSBMA/BNC2); tensile strength (43 ± 7 MPa for PSBMA/BNC1 and 28 ± 4 MPa for PSBMA/BNC2); elongation at break (0.7 ± 0.2% for PSBMA/BNC1 and 0.8 ± 0.4% for PSBMA/BNC2); high water-uptake capacity (450–559%); and UV-blocking properties. The PSBMA/BNC1 film exhibited bactericidal activity against *S. aureus* and *E. coli* with 1.3–log CFU/mL and 0.6–log CFU/mL reductions, respectively, whereas the PSBMA/BNC2 film displayed bactericidal activity against *S. aureus* and *E. coli* with 4.3–log CFU/mL and 1.1–log CFU/mL reductions, respectively [92].

#### 6.2.2. Chitosan Nanofibers

Lin et al., developed antibacterial packaging materials that were composed of chrysanthemum essential oil (CHEO)-loaded chitosan nanofibers through the electrospinning method. The antibacterial activity of CHEO demonstrated a MIC value of 2.5 mg/mL against *L. monocytogenes*. The antibacterial application of the CHEO nanofibers against *L. monocytogenes* was tested on beef, with an inhibition rate of 99.91%, 99.97%, and 99.95% at the temperature of 4 °C, 12 °C and 25 °C, respectively, after 7 days of storage. The antioxidant components in the CHEO that was released from the CHEO/CS nanofibers reduced the thiobarbituric acid reactive substances value in treated beef (0.135 MDA/kg), compared to the untreated sample that was stored at 4 °C after 12 days. The antimicrobial and antioxidant activities were attributed to CHEO in the nanofiber packaging material [93].

#### 6.2.3. Gelatin Nanofibers

Lin et al., fabricated moringa oil (20 mg/mL)-loaded chitosan nanoparticles (3.0 mg/mL) (MO@CNPs)-embedded gelatin nanofibers for the biocontrol of *L. monocytogenes* and *S. aureus* on cheese. The nanoparticle exhibited a desirable particle size, PDI and zeta potential ranging from 94.3 ± 2.1 to 246.1 ± 6.3 nm; 0.139 ± 0.017 to 0.432 ± 0.029; and 17.2 ± 1.5 to 45.1 ± 4.2 mV respectively. Furthermore, MO@CNPs (9.0 mg/mL) that were embedded in gelatin nanofibers exhibited desirable physical properties such as thickness (0.113 ± 0.002 mm); moisture content (13.42 ± 0.17%); water solubility (86.13 ± 0.15%); WVP (0.36 ± 0.05 g·mm/m^2^·h·kPa); tensile strength (1.24 ± 0.34 MPa); and elongation at break (47.61 ± 2.84%). MO@CNPs nanofibers exhibited high antibacterial activity against *L. monocytogenes* and *S. aureus* at 4 °C (percent inhibition 78.63% and 98.67% respectively) and 25 °C (3.11 and 2.2 Log CFU/g respectively) for 10 days when used on cheese, with no impact on the sensory quality of the cheese. The encapsulation of the nanofibers resulted in the controlled release of moringa oil from the nanoparticles, thus providing a prolonged shelf life [94].

#### 6.2.4. Polysaccharide Nanofibers

Cui et al., developed active food packaging containing Phlorotannin (PT) as antibacterial and antioxidant agent-encapsulated in *Momordicacharantia* polysaccharide (MCP) as a nanofiber matrix with an average diameter of 72.8 nm. The developed nanofibers demonstrated physical and mechanical properties such as thickness, moisture content, and water solubility and were found be in the range of 0.027 to 0.063 mm; 4.28 to 8.91%; and 10.42 to 18.94%, respectively. MCP exhibited MIC value of 138 mg/mL and MBC value of 254 mg/mL against *E. coli* O157, while PT demonstrated better antibacterial action against *E. coli* O157 than MCP, with lower MIC and MBC values (16 mg/mL and 64 mg/mL, respectively) and showed diameter of inhibition zone of 21.5 mm. Moreover, the antioxidant activity of MCP was found to be 90.35% and that of PT was found to be between 70–90% [95].

#### 6.2.5. Synthetic Biopolymeric Nanofibers

Radusin et al., developed active films of polylactic acid fibers containing extract of *Allium ursinum* L. (AU) (10 wt.%) in the 1–2 μm range with a bead-like shape and a mean diameter of 1868 ± 388 nm, by the electrospinning technology. The tensile strength at yield and elongation at break for the neat PLA sheet were 2.68 ± 0.43 MPa and 4.25 ± 0.83%, respectively, and on addition of the AU extract the value of tensile strength enhanced at yield to 4.76 ± 0.58 MPa while the elongation at break decreased to 3.48 ± 0.67%. The antimicrobial activity of the electrospun AU-containing polylactide film against *E. coli* and *S. aureus* bacteria exhibited an inhibition growth of 73.0 ± 0.85% and 27.4 ± 0.57% respectively [96].

Lan et al., fabricated PVA fibers containing D-limonene by electrospinning method for antimicrobial active packaging applications. The results of the mechanical properties showed that the highest tensile strength of 3.87 ± 0.25 MPa and elongation at break of 55.62 ± 2.93% were achieved for a PVA/D-limonene ratio of 7:3 (*v/v*) and an ultrasonication time of 15 min during processing. In addition, this material had the least oxygen permeability and the best degradability of all the samples. PVA/D-limonene ratio of 7:3 (*v/v*) also exhibited the highest antimicrobial activity against *E. coli* and *S. aureus* of 65 ± 2.11% and 58 ± 3.28%, respectively [97].

Pan et al., fabricated polyvinyl alcohol/cinnamon essential oil/β-cyclodextrin (CPVA-CEO-β-CD)-nanofibrous films for the sustained release of antimicrobial essential oil by crosslinked electrospinning method. CPVA-0.5CEO-β-CD, CPVA-1.0CEO-β-CD and CPVA-1.5CEO-β-CD films exhibited in vitro antibacterial activity with a bacteriostatic diameter of 10.11 ± 1.17 mm, 10.45 ± 2.21 mm and 11.24 ± 2.12 mm, respectively, against *S. aureus* and 9.16 ± 2.34 mm, 10.21 ± 1.31 mm and 11.32 ± 2.10 mm, respectively, against *E. coli*. Furthermore, CPVA/β-CD/CEO nanofibrous films delayed the decay of mushroom during storage, indicating their potential implementation in active food packaging [85].

Min et al., developed pullulan (PUL)/polyvinyl alcohol nanofibers that were incorporated with thymol-loaded porphyrin metal-organic framework nanoparticles (THY@PMO-224 NPs) for antibacterial food packaging. PMO-224 had a loading capacity of around 20% for thymol. The mechanical breaking elongation of THY@PMO/PUL/PVA nanofibers (20.20%) was reduced but the tensile strength was enhanced (2.63 MPa). Under light irradiation, the THY@PMO/PUL/PVA nanofibers showed synergistic antibacterial activity against *E. coli* (99%) and *S. aureus* (98%). The biosafety polymeric film was shown in cell viability testing and fruit preservation research. The findings suggested that this unique nanofiber could have use in food packaging [98].

Rashidi et al., developed bioactive nanofibers made of ethyl cellulose, soy protein isolated and bitter orange peel extract (BOPE) by electrospinning technology. The nanofibers exhibited adequate thermal stability; increased porosity of 78%; maximum water vapor transfer rate of 657 g/m^2^·24; higher tensile stress of 6.12 MPa; and an average water contact angle of 82.3°. The greatest concentration (20 wt.%) of BOPE improved the antioxidant activity of nanofibers by 64.7% and inhibited the growth of *S. areus*, and *E. coli*. Thus, the result suggested that BOPE demonstrated both antimicrobial and antioxidant activity in the packaging system [99].

He et al., fabricated pomegranate peel extract (PPE) and sodium dehydroacetate (SD) incorporated polyvinyl alcohol-composite film by electrospinning. The film exhibited physical properties such as thickness, tensile strength, elongation at breaking, water content, water vapor permeability, and water vapor transmission rate that were found to be in the range of 0.040 ± 0.009 to 0.055 ± 0.003 mm; 2.42 ± 0.23 to 10.38 ± 0.23 MPa; 18.20 ± 0.94 to 181.85 ± 1.02%; 10.2 ± 0.19 to 19.3 ± 0.83%; 3.66 ± 0.02 to 4.60 ± 0.04 × 10^−11^ g/m·s·Pa; and 1.90 ± 0.03 to 2.47 ± 0.02 × 10^−3^ g/m^2^·s. The antibacterial test findings demonstrate that the PVA/PPE/SD composite film showed strong antibacterial activity against *E. coli* and *S. aureus*, with *S. aureus* having a better antibacterial impact than *E. coli*. PPE and SD exhibited synergistic antibacterial action when combined, resulting in the highest inhibition rate of 96% and 93% against *S. aureus* and *E. coli*, respectively [100].

Amongst the nanofibers discussed above, cellulose nanofibers are extensively used for food packaging as they form great polymeric matrices that can be used as carriers for antimicrobial agents. Chitosan nanofibers show antimicrobial action along with acting as a carrier for other antimicrobial agents. Other synthetic biopolymeric nanofibers such as PVA, PLA, ethyl cellulose, etc., are widely used polymers in food packaging. A summary of all nanomaterials and antimicrobial agent incorporated in them along with efficiency against foodborne pathogens are given in Table 2.

## 7. Smart Packaging

Freshness of food is mainly indicated by a visual change, such as color. As previously stated, some active food packaging materials can also be used as sensors for determining the freshness of food. Typically, the color shift occurs as a result of biochemical processes in the food itself. A pH shift or the release of a specific molecule occurs as a result of the food’s progressive breakdown over time, but similar changes can also occur as a result of rapid heat changes. Consumers can immediately see the color change in the studies given below. If the price is affordable, such smart, sensitive food packaging materials can be immensely appealing to consumers [101].

Qin et al., developed starch/polyvinyl alcohol-based active and intelligent packaging sheets containing red pitaya (*Hylocereus polyrhizus*) peel extract (0.25, 0.50 and 1.00 wt.% on starch basis) that was rich in betalains. Betacyanins, the primary components of the extract, displayed substantial color changes when exposed to alkaline circumstances. The extract improved the mechanical, ultraviolet–visible light barrier, water vapor barrier, antioxidant, and antimicrobial capabilities of the films, with the best result obtained at 1.00 wt.%. The film with 1.00 wt.% extract was more ammonia-sensitive than the other films. The film containing 1.00 wt.% of the extract showed noticeable color changes due to the accumulated volatile nitrogen compounds during the deteriorating process of shrimp when used to check their freshness [102].

Duan et al., developed pullulan/chitin nanofibers (PCN) containing curcumin (CR) and anthocyanins (ATH) through the electrospinning technique for active-intelligent food packaging. The thickness, tensile strength, and elongation at break of the PCN/CR/ATH nanofiber was found to be 0.48 ± 0.02 mm, 10.18 ± 4.37 MPa and 10.05 ± 6.83%, respectively. The antioxidant activity (DPPH free radical scavenging rate) of the PCN/CR/ATH nanofiber was found to be 61.72% ± 1.73%. The nanofibers exhibited excellent antimicrobial activity against *S. aureus* and *E. coli* with an inhibition zone of 22.67 ± 0.76 mm and 22.83 ± 0.58 mm, respectively. The color of PCN/CR/ATH nanofibers changed significantly with the change in pH i.e., it appeared emerald-green at pH 8, dark green at pH 9 and 10, and dark yellow–green at pH 11, indicating the pH sensitivity of the nanofibers. The PCN/CR/ATH nanofibers clearly changed color from pink to powder blue with the progressive spoilage of *Plectorhynchuscinctus* at room temperature during 72 h [103].

Liu et al., developed an intelligent starch/poly-vinyl alcohol film that was loaded with anthocyanin (ANT) and limonene (LIM) and was capable of monitoring pH changes and inhibiting undesired microbial growth in foods. The results of the mechanical strength test showed that starch-PVA-ANT-LIM possesses the highest mechanical strength of 4.51 ± 0.14 MPa and its elongation at break was found to be 11.39 ± 0.12%. The films exhibited good antimicrobial properties against *Bacillus subtilis* (7 CFU/mL), *Staphylococcus aureus* (24 CFU/mL) and *Aspergillus niger* (4 CFU/mL). There was a distinct alteration of colors as the film was immersed in solutions of pH ranging from 1.0 (red-orange) to 14.0 (intense blue-green color). Finally, the film demonstrated good color indication and antimicrobial activity on pasteurized milk [104].

## 8. Edible Films

In the active food packaging industry, edible packaging is viewed as a sustainable and biodegradable alternative that improves food quality when compared to conventional packaging. The value of edible packaging can be demonstrated in its ability to preserve food quality, prolong shelf life, reduce waste, and contribute to packing material efficiency. Due to their versatility, potential to be manufactured from a range of materials, and ability to carry various active compounds, such as antioxidants and/or antibacterial agents, edible films are one of the most promising disciplines in food science. The food packaging materials are made from edible materials, such as natural polymers, which may be consumed by humans without posing any health risks. These materials can be changed into different types of films and coatings by varying their thicknesses rather than changing their material composition. Wraps, pouches, bags, capsules, and casings are commonly made with films, whereas coatings are placed directly onto the food surface. The coatings, in contrast to the films, are regarded as an intrinsic element of the food product and are normally not designed to be removed. Therefore, the right selection of edible packaging components is primarily determined by the food product to be packaged and the nature of the material that is used to create the edible packaging, as well as the processing method. Likewise, the packaging must be sensory-compatible with the food [105].

Chakravartula et al., developed cassava starch/chitosan (CS/CH) blends with pitanga (*Eugenia uniflora* L.) leaf extract (PE) and/or natamycin (NA) by casting method as active films for food packaging. The films with actives incorporated showed good UV barrier, ~18% to ~12% moisture content and mechanical strength. The addition of additives enhanced the radical scavenging activity of ABTS and DPPH radicals from 7.68% and 3.38% to 59.88% and 86.20%, respectively. Films containing just NA showed inhibitory zones against *A. flavus* (10.5 ± 0.7 mm) and *A. parasiticus* (11.0 ± 0.1 mm) in terms of antifungal activity, whereas films with PE and NA showed 11.0 ± 0.1 mm inhibition for *A. parasiticus* and 6.5 ± 0.7 mm inhibition for *A. flavus*, indicating antagonistic effects. Other films had no influence on mold growth [106].

Moghimi et al., fabricated edible hydroxyl propyl methyl cellulose films containing nanoemulsions of *Thymus daenensis* EO for food preservation. The edible film demonstrated thickness in the range from 239.2 ± 2.4 to 233.4 ± 4.1 µm; tensile strength in the range from 19.3 ± 1.0 to 22.6 ± 0.7 MPa; Young’s Modulus in the range from 62.5 ± 0.5 to 64.0 ± 1.6 MPa; and elongation at break in the range from 9.02 ± 0.3 to 14.2 ± 0.04 cm. The edible films containing EO exhibited significant antimicrobial activity against *S. aureus* with an inhibition zone of 47.0 ± 2.5 mm and 22.6 ± 0.5 mm, respectively. Thus, incorporation of EO nanoemulsions into the HPMC films can be used for active food preservation [107].

Göksen et al., developed active edible electrospun coatings containing *Laurus nobilis* essential oil (LEO) and *Rosmarinus officinalis* essential oil (REO) incorporated within zein matrix for active food packaging applications. The results demonstrated that the developed nanofiber (ZNF) films exhibited significant antimicrobial activity with MIC values of 0.417 ± 0.03 mg/mL for 5% LEO; 0.211 ± 0.04 mg/mL for 10% LEO; 0.640 ± 0.04 mg/mL for 5% REO; and 0.332 ± 0.01 mg/mL for 10% REO against *S. aureus*, while MIC values of 5% and 10% LEO and REO against *L. monoctogenes* were found to be 0.359 ± 0.01 mg/mL; 0.162 ± 0.01 mg/mL; 0.501 ± 0.02 mg/mL; and 0.273 ± 0.01 mg/mL, respectively,. The EOs-zein nanofibers (ZNF) films when coated on an inoculated side of cheese slices were found to be more effective (2–3 logs reduction) against mesophilic bacteria than for *L. monocytogenes* or *S. aureus* strains. *Rosmarinus officinalis* EO was found to exhibit stronger antibacterial activity than *Laurus nobilis* EO. After 28 days at 4 °C, the antibacterial efficiency of the active films increased, with a substantial reduction of 2 logarithm units of *L. monocytogenes* and *S. aureus* compared to the control samples [108].

Gebrechristos et al., fabricated potato starch films that were incorporated with potato peel extract to produce active edible film for food packaging. The film exhibited antimicrobial activity against *E. coli*, *S. aureus*, and *S. enterica* with minimum inhibitory concentrations of 7.5 ± 2, 4.7 ± 1 and 5.8 ± 2 mg/mL, respectively, but negative results in *L. monocytogenes* and *K. pneumoniae*. The scavenging activity and phenolic content of active film vary from 10 to 22 mg GAE/g dry film, with 24% to 55% inhibition, respectively. The potato peel extract exhibited antibacterial and antioxidant properties in the active film [109].

Atta et al., developed edible and bioactive films with yeast (*Meyerozyma guilliermondii* and *Gluconacetobacte rxylinus*) that was inserted into bacterial cellulose (BC) in conjunction with carboxymethyl cellulose (CMC) and glycerol (Gly) to extend the shelf life of packaged food products. The physical and mechanical properties such as tensile strength, elongation at break, moisture content and water solubility of the film were found to be 2.23 ± 0.33 MPa; 15.53 ± 0.84%; 23.66 ± 1.59%; and 42.86 ± 2.78%, respectively. After 24 h, the films that were incorporated with yeast demonstrated antimicrobial activity against three bacteria strains: *P. aeruginosa*, *E. coli* and *S. aureus*, with distinct inhibition zones of 10 mm, 16 mm and 15 mm, respectively. The films were also non-toxic to NIH-3T3 fibroblast cells. For up to two weeks, oranges and tomatoes that were coated with BC/CMC/Gly/yeast composite film displayed acceptable sensory qualities such as odor and color at 6°C, room temperature, and increased temperatures of 30 °C and 40 °C [110].

Sabaghi et al., developed chitosan coatings containing green tea extract (GTE). The CS10 covering that was mixed with GTE effectively inhibited lipid oxidation and fungal growth during walnut kernel storage (18 weeks). GTE proportions had no influence on lipid oxidation. Growth of fungi such as yeasts and molds were not observed during the period of storage with CS10 and all different proportions of GTE. Coatings without GTE had no significant effect on sensory qualities during storage whileCS10-GTE10 was found to be highly unsatisfactory. The results suggested that CS10-GTE5 coating could aid to extend shelf life walnut kernels [111].

In packaged foods, the primary function of edible films and coatings is to provide an active function such as antimicrobial, antioxidant, or barrier protection. Furthermore, plant extracts and oils including thyme, tea tree, bay leaf essential oil, rosemary essential oil, and pitanga leaf extracts, when used in food packaging, provide a healthy alternative to traditional packaging due to their antimicrobial and antioxidant characteristics for both the food and the customers. Table 3 summarizes edible films with antimicrobial activity against foodborne microorganisms.

## 9. In-Field Food Packaging Applications

Agricultural food products, such as fruits and vegetables, and derived food products, such as bread, cheese and meat products, etc., are more susceptible to foodborne pathogens such as *L. monocytogenes*, *E. coli*, *S. typhimurium* and *S. aureus*. Nanomaterial food packaging was found to inhibit the disease-causing pathogens and prolong the shelf life of the food product.

Akbar et al., developed zinc oxide nanoparticles (100 nm)-loaded calcium alginate film for active packaging. Zinc oxide nanoparticles-loaded active films exhibited antibacterial activity against *S. typhimurium* and *S. aureus* with an inhibition zone ranging from 16.6 ± 0.68 to 29.4 ± 0.99mm and 17.0 ± 0.96 to 32.5 ± 0.50 mm respectively. Within 10 days of incubation at 81°C, the film was used in ready-to-eat poultry meat as active packaging against the same microorganisms, and it reduced the number of inoculated target bacteria from log seven to zero [112].

Kim et al., developed nanoemulsions of lemongrass oil (LO)-loaded carnauba-based solution for the coating of grape berries (*Vitis labruscana* Bailey) to extend the shelf life and ensure their microbiological safety. The coating of the berries with 3.0 g/100 g LO reduced the number of *E. coli* O157:H7 and *S. typhimurium* that were inoculated on the berries by more than 2.6 and 3.2 log CFU/g, respectively. On storage at 4 and 25 °C for 28 days, the coating effectively reduced the growth of *S. typhimurium* and *E. coli* O157:H7 on the berries. Grape berries with LO-nanoemulsion coatings were found to be resistant to foodborne disease contamination and have a longer shelf life [113].

Noshirvani et al., developed a chitosan-carboxymethyl cellulose-oleic acid (CMC-CH-OL) nanocomposite film and coating that was integrated with varying concentrations of zinc oxide nanoparticles (ZnO NPs) (0.5, 1 and 2%) as a packaging material to extend the shelf life of sliced wheat bread. Microbiological studies demonstrated that CMC-CH-OL-ZnO NPs 2% increased the microbial shelf life of sliced wheat bread from 3 to 35 days when compared to the control. All active coatings reduced the number of yeasts and molds in sliced bread over the course of 15 days, with a minimum count of 2.45 ± 0.04 log CFU/g, while coatings containing 1 and 2% ZnO NPs exhibited no fungal growth over the course of 15 days [114].

Cui et al., developed chitosan that was loaded with a nisin-silica liposome of 138.7 nm to 149.2 nm in size and investigated its antimicrobial effects on Cheddar cheese. Nisin was allowed to be adsorbed on the surface of silica particle which increased the encapsulation efficiency of nisin in a liposome significantly from 65.5 to 75.7%. Nisin-silica liposomes-loaded chitosan coatings exhibited a prolonged antibacterial activity against *L. monocytogenes* without affecting the sensory qualities of the cheese. In the model cheese suspension that was stored at 25 °C on day 7, the antibacterial activity of the nisin-silica liposomes against *L. monocytogenes* was 5.48 ± 0.43 log CFU/g, and it was 4.48 ± 0.22 log CFU/g on day 14 when stored at 4 °C. After 7 days of storage at 25 °C, there was no growth of *L. monocytogenes*, implying that Nisin-silica liposomes-loaded chitosan coatings could be a possible active antibacterial for cheese preservation [115].

## 10. Safety, Innocuity and Toxicity of Nanomaterials

Aside from the numerous benefits of nanomaterials to the food industry, safety concerns about nanomaterials must not be overlooked. Many researchers explored nanomaterial safety concerns, focusing on the risk of nanoparticles migrating from packaging material into food and their influence on consumer health. Even if a substance is GRAS (generally regarded as safe), extra research is required to assess the risk of its nano equivalents since the physiochemical properties of nanostates differ significantly from those of macrostates. Furthermore, the small size of these nanomaterials may raise the danger of bioaccumulation within body organs and tissues [116].

Inhalation, ingestion, or cutaneous exposures are all possible routes for nanoparticles that are incorporated into nano-packed food products to enter the body. Since NPs are not soluble in biological fluids, they accumulate in organelles if they enter the circulation. The possible toxicity of NPs in food packaging has received very little attention. The migration of low molecular mass-nano packaging particles into food products is a major subject of concern for both scientists and customers. The concentration and particle size of NPs, and the nature of the food, play a major role in their migration to the food matrix. Temperature and acidity affect the migration of metal NPs into the food matrix. NPs have been shown to be genotoxic and carcinogenic in a few investigations [117].

The investigation of Ag migration revealed that acidic conditions accelerated Ag release from polymers [118]. Ag NPs can accumulate in different organs, including the testicles, kidneys, brain, and liver. In Sprague-Dawley adult rats, the oral administration of Ag nanoparticles at doses of 50, 100, and 200 mg/kg-day resulted in Ag bioaccumulate nanoparticles. Furthermore, a high dose of Ag nanoparticles in the body can be neurotoxic, hepatotoxic, and genotoxic [119]. The dissolution of nanoparticles of Zn in artificial lysosomal fluids (pH = 5.5) was higher than interstitial fluids (pH = 7.4), indicating that the migration of Zn nanoparticles is affected by the pH of the media [120]. CuNPs have been linked to cytotoxicity that is mediated by oxidative stress-related mechanisms. In HepG2 hepatocarcinoma and Caco-2 colorectal adenocarcinoma cell lines, CuONPs caused cytotoxicity, genotoxicity, and oxidative and apoptotic effects [121]. Human erythrocytes, murine fibroblasts (NIH-3T3), human cervical cancer cells (HeLa), and melanoma cells (B16F10) were all tested for AuNPs cytotoxicity. The cytotoxic effect of AuNPs appears to be limited by the physicochemical features and concentration of AuNPs, and the cell type [122]. TiO_2_NPs are cytotoxic or genotoxic, while low dosages of nano TiO_2_ appear to be non-toxic [123]. The human intestine Caco-2 cell line has harmful effects because silica NPs permeates the cytoplasm but not the nucleus. They hypothesized that silica genotoxic effects of NPs are more likely to be mediated by oxidative stress rather than direct contact with DNA [124].

The shape, size, size distribution, structure, composition, surface functionality, porosity, surface area, surface charge, aggregation, concentration, and solubility of NPs have all been found to play a role in their toxicity in investigations. The biological and pathological consequences of NPs should be assessed by several characteristics, including NPs’ physiochemical properties, concentration, dose, exposure route, and duration, in order to assess their toxicity. The size of NPs, the dose, and the exposure time appear to have a significant impact on their toxicity. The current data are based on cell culture research in vitro and/or animal model investigations in vivo. The effectiveness of these models in predicting NP toxicity in humans is debatable. Existing models must be used with caution when investigating and understanding the biological and pathophysiological mechanisms of NP toxicity [46].

## 11. Commercialized Nanomaterial Food Packaging

Commercial nanoclay and nanosilver materials are widely used for food packaging. These materials enhance the physical and mechanical properties, oxygen barrier, antioxidant and antimicrobial properties of the food packages. In 2014, the global nanoclay market for food packaging was the largest category, with USD 343M in revenue, and it is predicted to continue to develop significantly in the future [125]. This surge in the use of nanoclays could be due to their large availability, natural occurrence, low cost, and ecological appeal. They improve the mechanical and barrier qualities of packaging materials by increasing the tortuosity of a penetrant molecule’s diffusive path, forcing them to travel a longer distance to diffuse through the matrix, and they could be used as carrier of antimicrobial substance [126]. Nanosilver has a low toxicity as compared to other metal nanoparticles which makes them a suitable choice for commercial food packages. Moreover, silver has antimicrobial properties and can be easily incorporated into polymers that are used for packaging [127]. Table 4 summarizes the commercial products of nanoclay and nanosilver with their effect on pathogens, environment and human safety.

## 12. Conclusions

Foodborne pathogenic microbes (bacteria, parasites, and viruses) have a strong potential to alter the appearance, taste, and quality of food products and can contaminate food, resulting in foodborne diseases. Conventional food packaging systems provide only physical support and protection against the environments that are encountered during the packaging process, distribution, transportation, and storage. These systems are unable to meet the demands of the current consumer. This has led to an upsurge in the development of improved preservation technology and innovative packaging to prevent foodborne pathogen contamination. Nanotechnology provides a platform to develop novel food packaging nanomaterials with unique physiochemical and antimicrobial attributes. It helps in the utilization of preservatives and antimicrobial agents to extend the shelf life of food within the package. Advances in antimicrobial nanomaterials and their application in food packaging materials have revolutionized food preservation, storage, distribution, and the consumption of packaged food. Antimicrobial nano packaging includes improved packaging, active packaging and intelligent/smart packaging types which all have a unique basis and material for preserving the quality of packaged food. They utilize antimicrobial nanomaterials such as antimicrobial inorganic metallic and metal oxide nanoparticles (silver, gold, copper and titanium dioxide, silicon oxide, and zinc oxide) that provide superior food safety and shelf life to packaged foods.

The evolving landscape of nanomaterials includes nanocomposites, nanoparticles, nano emulsions, nanoliposomes, nano sponges and nanofibers and these materials curb microbial invasion, promoting food safety. In addition, smart packaging incorporates nanosensors that alert and warn consumers about the safety and accurate nutritional status of the packaged food. Smart packaging that is incorporated with indicators such as betalain, anthocyanin, and other nature-derived compounds that exhibit color changes in response to a change in pH (due to spoilage or microbial contamination of food) are attractive options to the consumer. Edible films and coatings incorporated with plant extracts and oils including thyme, tea tree, bay leaf essential oil, rosemary essential oil, and pitanga leaf extract are reported to be beneficial in these packaging systems and help in monitoring changes in the internal and external environment of the food package. Despite theseveral advantages of nanomaterials-based food packaging systems, there are limited reports on their safety and innocuity, warranting further research to make these smart packaging systems universal in the food industry.

## Figures and Tables

**Figure 1 antibiotics-11-00729-f001:**
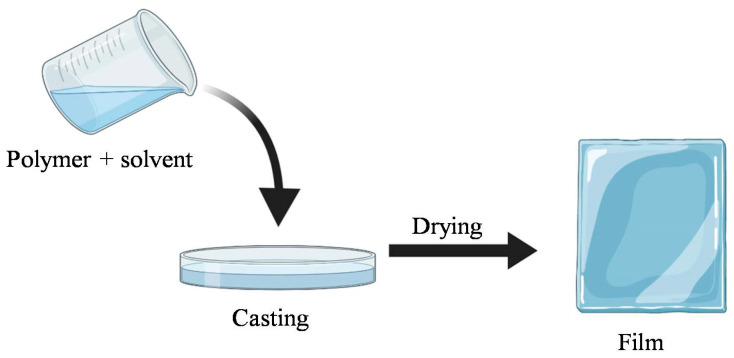
Casting method.

**Figure 2 antibiotics-11-00729-f002:**
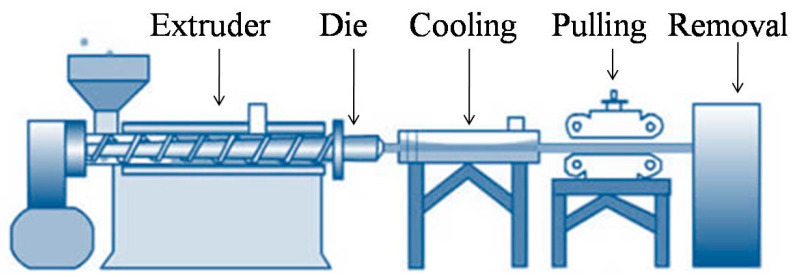
Film extrusion process.

**Figure 3 antibiotics-11-00729-f003:**
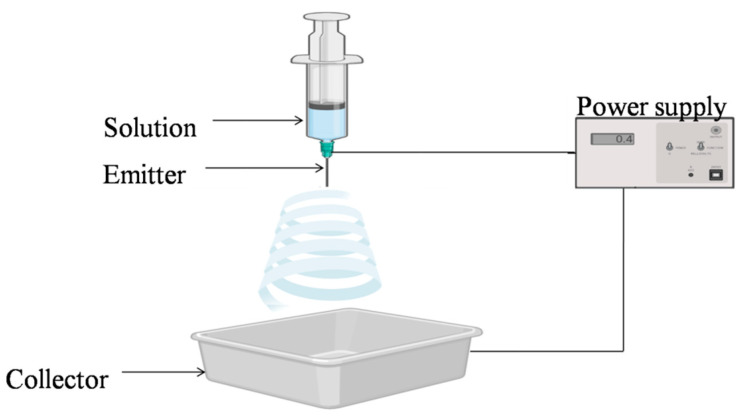
Electrospinning method.

**Table 1 antibiotics-11-00729-t001:** Foodborne pathogens, their effect on humans, and available treatments.

Bacterial Strains	Food Vehicle	Effects	Treatments	References
*Escherichia coli* O157:H7	Contaminated hamburger meat, unpasteurized milk, tomatoes, white radish sprouts, lettuce, fresh spinach, and apple juice	Non-bloody diarrhea, hemorrhagic colitis, hemolytic uremic syndrome, thrombocytopenia purpura, and fatality	Azithromycin, rifampicin and gentamicin	[16]
*Campylobacters* sp.	Raw milk, raw red meat, fruits and vegetables	Fever, stomach pain, vomiting, and dehydration, watery stools containing leukocytes, Guillain-Barré Syndrome (GBS), Reactive Arthritis (REA), and irritable bowel syndrome.	Tetracycline, ciprofloxacin, fluoroquinolones and erythromycin	[17]
*Shigella*	Tomatoes, ground beef, raw carrots, raw oysters, and bean salad	Dysentery or severe colitis, pseudo-membranous colitis, toxic megacolon, hemolytic uremic syndrome, intestinal perforation, septicemia, and convulsions	Fluoroquinolones (first-line), β-lactams (second-line), and cephalosporins (second-line)	[18]
*Staphylococcus aureus*	Meat and meat products, poultry and egg products, milk and dairy products, salads, bakery products (especially cream-filled pastries and cakes), and sandwich fillings	Hypersalivation, nausea, vomiting, and abdominal cramping with or without diarrhea	β-lactams, oxacillin, nafcillin, cefazolin, vancomycin, daptomycin, linezolid, quinupristin/dalfopristin, cotrimoxazole, ceftaroline, telavancin etc.	[19,20]
*Listeria monocytogenes*	Milk and milk products such as cheese, ice cream, butter, cream, yogurt, etc	Diarrhea, mild fever, nausea, and vomiting	Ampicillin, ceftriaxone, cephalothin, clindamycin, gentamicin, meticillin, oxacillin, streptomycin etc.	[21]
*Salmonella typhimurium*	Poultry, beef, egg, and dairy products	Gastric carriage, gastroenteritis, bacteremia, meningitis, and osteomyelitis	First-line antibiotics ciprofloxacin and ceftriaxone	[17]

**Table 2 antibiotics-11-00729-t002:** Antimicrobial nanoparticles against foodborne pathogens.

Nanoparticles	Antimicrobial Agents	Pathogens	Efficiency	Reference
Cellulose nanocrystals	-	*S. aureus*, *E. coli* and *C. albicans*	100% inhibition	[90]
	Tea polyphenol	*E. coli* and *S. aureus*	ZOI—~12 mm and ~16 mm respectively	[91]
	Nisin	*L. monocytogenes*	100% inhibition	[88]
Chitosan	Moringa oil	*L. monocytogenes* and *S. aureus*	Exhibited high antibacterial activity at 4 °C (percent inhibition 78.63% and 98.67% respectively) and 25 °C (3.11 and 2.2 Log CFU/g respectively) for 10 days	[94]
Copper	-	*S. aureus*, *Salmonella* sp., *C. albicans*, *E. coli*, and *Trichoderma* sp.	ZOI of 27.49 ± 0.91 mm, 25.21 ± 1.05 mm, 23.35 ± 0.45 mm, 12.12 ± 0.58 mm, and 5.31 ± 1.16 mm, respectively	[86]
Gold	-	*E. coli*	ZOI—10 mm	[54]
Graphitic carbon nitride	-	*E. coli* and *S. aureus*	99.8 ± 0.26% and 99.9 ± 0.04%, respectively	[56]
Graphite carbon nitride nanosheets/Molybdenum sulfide nanodots	Konjac glucomannan	*S. aureus* and *E. coli*	ZOI—~2.1 cm and ~1.3 cm, respectively	[55]
Halloysite nanotubes	Thyme essential oil	*E. coli*	Reduced bacterial count to ~2.5 log CFU/cm^2^	[53]
Magnesium oxide	-	*L. monocytogenes* and *S. baltica*	99.99% inhibition	[57]
Palladium and platinum	-	*E. coli*, *S. entericaInfantis*, *L. monocytogenes*, and *S. aureus*	0.3–2.4 log CFU/mL (PdNPs) and 0.8–2.0 log CFU/mL (PtNPs), respectively	[58]
Silica/nanoclay/ montmorillonite	Cinnamon essential oil	*Mucor* species and *Mucor circinelloide s* strain	MIC—6 mg/mL	[62]
	Curcumin	*E. coli* and *S. aureus*	ZOI—~7.5 mm and ~8 mm, respectively	[64]
	Potassium sorbate/grapefruit seed extract	*Aspergillus niger*	ZOI—13.47 ± 0.79 mm to 47.10 ± 0.50 mm	[65]
	Silver	*S. aureus*, *E. coli, Salmonella*, and *P. aeruginosa*	100% inhibition	[66]
	Silver and thymol	*E. coli*, *S. aureus*, *Salmonella* sp., *Pseudomonas*, *A. niger*, and *A. flavus*	ZOI—28 ± 0.8 mm, 25 ± 0.5 mm, 20 ± 0.2 mm, 25.5 ± 1 mm, 22.5 ± 0.6 mm, 19 ± 0.2 mm, respectively	[67]
Silver	-	*S. aureus*, *E. faecalis*, *E. coli*, *S. typhimurium*, and *P. expansuma*	2.035 to 1.682 log CFU/mL, 1.493 to 0.934 log CFU/mL, 2.072 to 0.279 log CFU/mL, 1.625 to <1 log CFU/mL and <1 log CFU/mL, respectively	[70]
	Laurel essential oil	*S. aureus* and *E. coli*	ZOI—~5 mm and ~1 mm, respectively	[82]
Titanium dioxide	Rosemary oil	*L. monocytogenes* and *S. aureus*	4.15 logCFU/g	[83]
Zein colloid	Thymol	*E. coli* and *Salmonella*	ZOI—15.89 ± 0.74 mm to 18.81 ± 0.56 mm, respectively	[80]
Zinc oxide-chitosan	Antioxidant of bamboo leaves	*S. aureus* and *E. coli*	ZOI—27.01 ± 1.28 to 28.54 ± 3.55 mm and 26.60 ± 3.00 to 29.69 ± 2.53 mm, respectively	[74]
	-	*S. aureus* and *E. coli*	100% and 65% inhibition	[73]
ZnO-Silver	-	*E. coli* and *S. aureus*	ZOI—~12–22 mm and ~14–25 mm, respectively	[77]

**Table 3 antibiotics-11-00729-t003:** Antimicrobial edible films against foodborne microorganisms.

Packaging Material	Antimicrobial Agent	Microorganisms	Efficiency	Reference
Cassava/starch/chitosan	Pitanga leaf extract and/or natamycin	*A. flavus* and *A. parasiticus*	ZOI—11.0 ± 0.1 mm and 6.5 ± 0.7 mm	[106]
Hydroxyl propyl methyl cellulose	*Thymus daenensis* EO	*S. aureus*	ZOI—47.0 ± 2.5 mm and 22.6 ± 0.5 mm	[107]
Zein matrix	*Laurus nobilis* essential oil (LEO) and *Rosmarinus officinalis* essential oil (REO)	Mesophilic bacteria, *L. monocytogenes* and *S. aureus*	MIC (LEO) 0.211 ± 0.04 mg/mL and MIC (REO) 0.162 ± 0.01 mg/mL	[108]
Potato starch	Potato peel extract	*E. coli*, *S. aureus*, and *S. enterica*	MIC values of 7.5 ± 2, 4.7 ± 1 and 5.8 ± 2 mg/mL, respectively	[109]
Bacterial cellulose (BC) combined with carboxymethyl cellulose	Glycerol	*P. aeruginosa*, *E. coli* and *S. aureus*	ZOI—10 mm, 16 mm and 15 mm, respectively	[110]
Chitosan	Green tea extract	Yeasts and molds	-	[111]

**Table 4 antibiotics-11-00729-t004:** Commercial products of nanoparticles in food packaging with their effect on pathogens, environment and human safety.

Nanoparticle	Marketed Product	Pathogens	Environmental Safety	Human Safety	References
Nanoclay	Aegis™ OXCE, Durethan^®^ KU2-2601, Imperm^®^, Cloisite^®^, Nanocor^®^, Nanolin^®^, Dellite^®^, Shelsiteand Plantic^®^	*E. coli*, *S. aureus*, *Salmonella* and *Aspergillus niger*	Generally recognized as safe (GRAS)	Lung diseases, genotoxicity and platelet thickening observed workers exposed	[128,129,130]
Nanosilver	FresherLonger™, Sina, e.Window^®^, Everin, Incense, Fresh Box, BabyDream, Zeomic, Anson, and Dokdo	*S. aureus*, *E. faecalis*, *E. coli*, *S. typhimurium*, and *P. expansuma*	Toxicity at high concentration (above 10 mg/Kg)	Bioaccumulation in various organs such as testicles, kidneys, brain, and liver- causing neurotoxic, hepatotoxic, and genotoxic effects	[119,128]

## Data Availability

Not applicable.

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
