# Peer review of "Antimicrobial Nanomaterials for Food Packaging"

_antibiotics, 2022, doi:10.3390/antibiotics11060729_

Round 1

Reviewer 1 Report

Review of the manuscript “Antimicrobial Nanomaterials for Food Packaging” by V. Suvarna, A. Nair, R. Mallya, T. Khan and A. Omri

This manuscript reviews the different nanomaterials used to produce active packaging with antibacterial properties and smart packaging that indicates the quality of the food.

In my opinion, the subject of the review is of great interest. I also think that that the authors have made a good selection of the papers for the review. Therefore, the  manuscript has potential to be considered for publication but it needs some major amendments that I shown below:

  • The paper need language polishing. Very long paragraphs should be avoided. In addition the text lacks of linking words that will conect one sentence with the following, making the manuscript more readable
  • There several sentences all over the text that are redundant. For example in line 135 the sentences “Imazalil is an antimycotic agent used 135 in food packaging. Imazalil(1-[2-(2,4-dichlorophenyl)-2-(2-propenyloxy)ethyl]-lH-imid-136 azole) is a fungal sterol biosynthesis inhibitor that is widely used to control a variety of 137 fungus” could be merged in one. For example, it can be written: “Imazalil(1-[2-(2,4-dichlorophenyl)-2-(2-propenyloxy)ethyl]-lH-imid-136 azole) is a fungal sterol biosynthesis inhibitor that is widely used to control a variety of  fungus in food packaging”.

Revise the text in order to eliminate similar redundant sentences (for example, sentences 339 to 343, can be simplified).

  • Line 70. PLA should be eliminated from section “a. Petroleum-based polymer materials” because it is a bio-based and biodegradable polymer. In fact, it is synthesized through the fermentation under controlled conditions of a carbohydrate source like corn starch or sugarcane.
  • Lines 140 to 163. I think that, as this text is about a nanocomposite film, it should not be in the section “1.2 Antimicrobial-agents overview”.
  • In line 378 the following is stated “Nanomaterials can be used as growth inhibitors, killing agents, or even antibiotic transporters in antimicrobial films”. This is correct, however, I think that in most of the examples of the Section “6 Antimicrobial Nanomaterial Used for Active Packaging” this is not clearly explained. In some of the examples it is not described if the nanomaterial has antibacterial activity “per se” or if it is a container of the antibacterial agent.
  • Line 404. I think that the sentence “The nanoparticles were coated onto paper to assess the 404 chemical and morphological features of coated sheets in terms of ink receipt, hydrophobicity, and optical qualities (gloss)” has not sense in the context of the paragraph.
  • Line 581, sentence “Antibacterial compounds can be encapsulated in electrospun nanofibers and used in fresh-keeping packaging” can be eliminated because the same is explained with the last sentence of the paragraph.
  • Section 11 (Conclusions) should be rewritten. It is a summary of the diferent sections but the sentences look not connected. In my opinion, there is a feeling of disorder. And also,
  • In my opinion, the main problem to fix appear in points “6.1. Nanoparticles” and “6.5. Nanocomposite Films”. I don´t see clear the criteria followed to include some of the examples in each sections.

Nanocomposite materials are formed of two or more phases that have different chemical and physical properties, and where one of the constituents has at least one dimension in the nanometer-size range. Based on the shape of the nanofillers, they are divided into nanoparticles, nanorods, nanotubes, nanowhiskers, or nanofibers, and are collectively termed nanostructures and can be broadly classified into organic and inorganic nanofillers.

Taking into account the above description of nanocomposite, the conclusion is that the examples of “6.1. Nanoparticles” are also nanocomposites. And examples of metallic nanoparticles, for example, ZnO, are included in section “6.1. Nanoparticles” and also in section “6.5. Nanocomposite Films”.

In summary, a clear criteria has to be selected because examples of section 6.1 and 6.2 are all nanocomposites. May be, a classification of nanocomposites based on composition of nanofillers can be made: metal, metal oxides, clays, biopolymers (cellulose derivatives, chitosan,..)

Author Response

Response to Reviewer 1

Review of the manuscript “Antimicrobial Nanomaterials for Food Packaging” by V. Suvarna, A. Nair, R. Mallya, T. Khan and A. Omri

This manuscript reviews the different nanomaterials used to produce active packaging with antibacterial properties and smart packaging that indicates the quality of the food.

In my opinion, the subject of the review is of great interest. I also think that that the authors have made a good selection of the papers for the review. Therefore, the  manuscript has potential to be considered for publication but it needs some major amendments that I shown below:

  • The paper need language polishing. Very long paragraphs should be avoided. In addition the text lacks of linking words that will conect one sentence with the following, making the manuscript more readable

Answer- Language correction and linking words are incorporated into the manuscript.

  • There several sentences all over the text that are redundant. For example in line 135 the sentences “Imazalil is an antimycotic agent used 135 in food packaging. Imazalil(1-[2-(2,4-dichlorophenyl)-2-(2-propenyloxy)ethyl]-lH-imid-136 azole) is a fungal sterol biosynthesis inhibitor that is widely used to control a variety of 137 fungus” could be merged in one. For example, it can be written: “Imazalil(1-[2-(2,4-dichlorophenyl)-2-(2-propenyloxy)ethyl]-lH-imid-136 azole) is a fungal sterol biosynthesis inhibitor that is widely used to control a variety of  fungus in food packaging”.

Answer- Recommended information has been incorporated into line no

Revise the text in order to eliminate similar redundant sentences (for example, sentences 339 to 343, can be simplified).

Answer- Simplification of line no done

  • Line 70. PLA should be eliminated from section “a. Petroleum-based polymer materials” because it is a bio-based and biodegradable polymer. In fact, it is synthesized through the fermentation under controlled conditions of a carbohydrate source like corn starch or sugarcane.

Answer- Recommended correction has been done into line no

  • Lines 140 to 163. I think that, as this text is about a nanocomposite film, it should not be in the section “1.2 Antimicrobial-agents overview”.

Answer- Recommended text has been shifted to section 6.5 as per the suggestion

  • In line 378 the following is stated “Nanomaterials can be used as growth inhibitors, killing agents, or even antibiotic transporters in antimicrobial films”. This is correct, however, I think that in most of the examples of the Section “6 Antimicrobial Nanomaterial Used for Active Packaging” this is not clearly explained. In some of the examples it is not described if the nanomaterial has antibacterial activity “per se” or if it is a container of the antibacterial agent.

Answer- Recommended information has been added as per the suggestion

  • Line 404. I think that the sentence “The nanoparticles were coated onto paper to assess the 404 chemical and morphological features of coated sheets in terms of ink receipt, hydrophobicity, and optical qualities (gloss)” has not sense in the context of the paragraph.

Answer- Recommended abrupt information has been deleted

  • Line 581, sentence “Antibacterial compounds can be encapsulated in electrospun nanofibers and used in fresh-keeping packaging” can be eliminated because the same is explained with the last sentence of the paragraph.

Answer- Recommended text has been deleted

  • Section 11 (Conclusions) should be rewritten. It is a summary of the diferent sections but the sentences look not connected. In my opinion, there is a feeling of disorder. And also,

Answer- Conclusion is rewritten

  • In my opinion, the main problem to fix appear in points “6.1. Nanoparticles” and “6.5. Nanocomposite Films”. I don´t see clear the criteria followed to include some of the examples in each sections.

Nanocomposite materials are formed of two or more phases that have different chemical and physical properties, and where one of the constituents has at least one dimension in the nanometer-size range. Based on the shape of the nanofillers, they are divided into nanoparticles, nanorods, nanotubes, nanowhiskers, or nanofibers, and are collectively termed nanostructures and can be broadly classified into organic and inorganic nanofillers.

Taking into account the above description of nanocomposite, the conclusion is that the examples of “6.1. Nanoparticles” are also nanocomposites. And examples of metallic nanoparticles, for example, ZnO, are included in section “6.1. Nanoparticles” and also in section “6.5. Nanocomposite Films”.

In summary, a clear criteria has to be selected because examples of section 6.1 and 6.2 are all nanocomposites. May be, a classification of nanocomposites based on composition of nanofillers can be made: metal, metal oxides, clays, biopolymers (cellulose derivatives, chitosan,..)

Answer- Recommended text has been rearranged

Reviewer 2 Report

In recent years, the development of food packaging systems has been gained much importance of the researchers and food industry. These systems play an important role in the food sector by helping to maintain the quality and prolong the shelf life of the food products. An ideal food packaging system can maintain the physicochemical attributes (e.g., color, flavor, moisture content, and texture) of foods and their raw materials. Further, they help to prevent oxidation and microbial deterioration of the food products. Antimicrobial food packaging systems are meant to protect the food from microbiological spoilage. The development of such antimicrobial systems employs various nanomaterials, which usually have preservative and antimicrobial action. The present review has excellently summarized the recent updates on nanomaterial-based food packaging technology. I have found the quality of the paper quite satisfactory.

However, I feel the authors have missed an important class of materials, namely lipid nanocarriers, which have also been explored in food packaging. If they can incorporate a section on the same, it would be great.  

Author Response

Response to Reviewer 2

Comments and Suggestions for Authors

In recent years, the development of food packaging systems has been gained much importance of the researchers and food industry. These systems play an important role in the food sector by helping to maintain the quality and prolong the shelf life of the food products. An ideal food packaging system can maintain the physicochemical attributes (e.g., color, flavor, moisture content, and texture) of foods and their raw materials. Further, they help to prevent oxidation and microbial deterioration of the food products. Antimicrobial food packaging systems are meant to protect the food from microbiological spoilage. The development of such antimicrobial systems employs various nanomaterials, which usually have preservative and antimicrobial action. The present review has excellently summarized the recent updates on nanomaterial-based food packaging technology. I have found the quality of the paper quite satisfactory.

However, I feel the authors have missed an important class of materials, namely lipid nanocarriers, which have also been explored in food packaging. If they can incorporate a section on the same, it would be great.  

Answer- Lipid nanocarriers has been included in section 6 as per the suggestion

Reviewer 3 Report

The review article antibiotics-1729431, entitled “Antimicrobial Nanomaterials for Food Packaging” addresses the most recent advances in the preparation and characterization of different nanomaterials for food packaging, focusing on antimicrobial, antioxidant and smart properties. Despite being well written, with a logical sequence of subdivisions, this review, in general, only reproduces the results found by other studies in the literature, without discussing them critically.

The paragraphs are very long, and the text could be reduced and much clearer if tables were used to summarize the studies cited.

Moreover, another critical problem in the article is the similarity index of this review with other studies or sources, which is 47%, according to the Turntin platform, a very high % for a review (please see attached).

In the case of review articles, it is interesting to critically discuss what other studies have already done, and not just repeat what has been done, which ends up leading to high similarities. I strongly encourage the authors to review entire text to reduce plagiarism. After the modifications in the text I can change my decision.

In addition, here are some specific comments. 

10-12: “Food packaging plays a key role in food industry to overcome the major obstacle encountered by the food sector in the supply of appetizing and accessible food with excellent quality and prolonged shelf life.” – the sentence is too long, please restructure it. 

24: “Recently intelligent food packaging…” – I see that the authors use both “smart/intelligent” terms through the manuscript; I suggest standardizing them, mentioning that they are synonymous only the first time they appear in the text, and then using only one term.

36: “…have recently been contaminated by pathogenic organisms due to a lack of safety practices.” – recently? When? Are the authors referring to a specific event? I suggest referencing it. 

57-58: “For example, as the pH rises from 2 to 10, the colour of a blueberry film changes from rose to blue-green.” – the citation (ref) is missing. 

58: “AP offers…” – please define “AP” as active packaging the first time that the term appears in the manuscript. 

63: did the authors take all information from “Petroleum-based plastic polymer materials” from reference 4 only? Citations of more studies are missing, which may be the cause of the high similarity index found in this manuscript. The same is valid for “Biodegradable polymers” and “Paper as packaging material” sections; they are too long for just one citation for each section. 

140: BHA/BHT: define the names of these preservatives the first time they appear in the manuscript. 

143-145: “Mondal et al., A solution mixing procedure was used to generate a poly(butylene 143 adipate-co-terephthalate) (PBAT)/ cetyltrimethylammonium-modified montmorillonite 144 (CMMT) based nanocomposite film doped with sodium benzoate (SB) as an antibacterial 145 agent” – the English is confusing, please restructure the sentence.

215: “Antimicrobial activity has been found in high-molecular-weight chitosan…” – as chitosan is the only biopolymer with antimicrobial action that the authors mention, I suggest exploring a little bit more about it. Not only the molecular weight interferes with chitosan’ antimicrobial action, but also its acetylation degree and source. 

222: “package” – standardize the use, whether it is “packaging” or “package”. The first term is most used. 

273: “The chosen polymer dissolved in suitable solution.” – I believe there is a verb missing in the sentence.

275: “mould” is fungus, I believe “mold” is the correct word. 

302: “During the drying process, the jet created fibres, which were then deposited on the collector.” – the sentence is confusing, restructure it. 

326-328: “Antimicrobial packaging solutions in general classified as migrating or non-migrating in general, with the distinction being made based on the antimicrobial agent utilized and its interactions with the packaging and food matrix.” – the sentence is confusing, please restructure it. 

Section 6: I suggest restructuring the whole section. There are problems with: (1) the paragraphs of each type of antimicrobial nanomaterial are too long, which makes the text tiring for the reader; (2) the information is only being reproduced, in relation to the results found in the different studies, and not critically analyzed; I suggest changing the main results found to a table (since there is none in the manuscript), divided by type of nanomaterial and with the references used, and leaving in the text only a critical assessment of what was found.

973: “Various nanomaterial used in active...” – I do not see why this paragraph fits here, at the end of section 6. It should come at the beginning of the nanocomposites discussion, for example. 

991: change “high” for “rich”.

1018: the name of bacteria must come in italics; standardize it through the whole text. 

1117: “Edible films and coatings are made from…” – again, I do not see this paragraph as the most suitable to close the section of edible films. It should come at the beginning of the section. Moreover, the paragraph before this one is too long and only reproduces the main findings of the studies, without discussing them. They should come summarized in a table. The same commentaries are all valid for section 9 (too long, does not critically say anything, the closing paragraph is confusing and in the wrong place).

Author Response

Response to Reviewer 3

The review article antibiotics-1729431, entitled “Antimicrobial Nanomaterials for Food Packaging” addresses the most recent advances in the preparation and characterization of different nanomaterials for food packaging, focusing on antimicrobial, antioxidant and smart properties. Despite being well written, with a logical sequence of subdivisions, this review, in general, only reproduces the results found by other studies in the literature, without discussing them critically.

The paragraphs are very long, and the text could be reduced and much clearer if tables were used to summarize the studies cited.

Answer- Text is reduced and condensed. Tables are incorporated as recommended.

Moreover, another critical problem in the article is the similarity index of this review with other studies or sources, which is 47%, according to the Turntin platform, a very high % for a review (please see attached).

Answer- Plagiarism report of revised manuscript according to Turntin platform is as attached.

In the case of review articles, it is interesting to critically discuss what other studies have already done, and not just repeat what has been done, which ends up leading to high similarities. I strongly encourage the authors to review entire text to reduce plagiarism. After the modifications in the text I can change my decision.

Answer- Entire manuscript is revised and plagiarism is reduced

In addition, here are some specific comments. 

10-12: “Food packaging plays a key role in food industry to overcome the major obstacle encountered by the food sector in the supply of appetizing and accessible food with excellent quality and prolonged shelf life.” – the sentence is too long, please restructure it. 

Answer- The sentence is restructured

24: “Recently intelligent food packaging…” – I see that the authors use both “smart/intelligent” terms through the manuscript; I suggest standardizing them, mentioning that they are synonymous only the first time they appear in the text, and then using only one term.

Answer- Recommended changes are incorporated

36: “…have recently been contaminated by pathogenic organisms due to a lack of safety practices.” – recently? When? Are the authors referring to a specific event? I suggest referencing it. 

Answer- Referencing is done and updated information is added

57-58: “For example, as the pH rises from 2 to 10, the colour of a blueberry film changes from rose to blue-green.” – the citation (ref) is missing. 

Answer- Suggested citation is added

58: “AP offers…” – please define “AP” as active packaging the first time that the term appears in the manuscript. 

Answer- Recommended changes are incorporated

63: did the authors take all information from “Petroleum-based plastic polymer materials” from reference 4 only? Citations of more studies are missing, which may be the cause of the high similarity index found in this manuscript. The same is valid for “Biodegradable polymers” and “Paper as packaging material” sections; they are too long for just one citation for each section.

Answer- Citations are added as per the suggestion

140: BHA/BHT: define the names of these preservatives the first time they appear in the manuscript. 

Answer- Full form of the abbreviation has been added

143-145: “Mondal et al., A solution mixing procedure was used to generate a poly(butylene 143 adipate-co-terephthalate) (PBAT)/ cetyltrimethylammonium-modified montmorillonite 144 (CMMT) based nanocomposite film doped with sodium benzoate (SB) as an antibacterial 145 agent” – the English is confusing, please restructure the sentence.

Answer- Restructuring of sentence is done

215: “Antimicrobial activity has been found in high-molecular-weight chitosan…” – as chitosan is the only biopolymer with antimicrobial action that the authors mention, I suggest exploring a little bit more about it. Not only the molecular weight interferes with chitosan’ antimicrobial action, but also its acetylation degree and source. 

Answer- Recommended information is added

222: “package” – standardize the use, whether it is “packaging” or “package”. The first term is most used. 

Answer- The ‘packaging’ word is retained

273: “The chosen polymer dissolved in suitable solution.” – I believe there is a verb missing in the sentence.

Answer- Correction is done

275: “mould” is fungus, I believe “mold” is the correct word. 

Answer- Correction is done

302: “During the drying process, the jet created fibres, which were then deposited on the collector.” – the sentence is confusing, restructure it. 

Answer- Restructuring is done

326-328: “Antimicrobial packaging solutions in general classified as migrating or non-migrating in general, with the distinction being made based on the antimicrobial agent utilized and its interactions with the packaging and food matrix.” – the sentence is confusing, please restructure it. 

Answer- Sentence is restructured as per the suggestion

Section 6: I suggest restructuring the whole section. There are problems with: (1) the paragraphs of each type of antimicrobial nanomaterial are too long, which makes the text tiring for the reader; (2) the information is only being reproduced, in relation to the results found in the different studies, and not critically analyzed; I suggest changing the main results found to a table (since there is none in the manuscript), divided by type of nanomaterial and with the references used, and leaving in the text only a critical assessment of what was found.

Answer- Section 6 has been restructured as per the suggestion

973: “Various nanomaterial used in active...” – I do not see why this paragraph fits here, at the end of section 6. It should come at the beginning of the nanocomposites discussion, for example. 

Answer- Changes are done as per the suggestion

991: change “high” for “rich”.

Answer- Changed to high as per the suggestion

1018: the name of bacteria must come in italics; standardize it through the whole text. 

Answer- Name bacteria are standardized to italics

1117: “Edible films and coatings are made from…” – again, I do not see this paragraph as the most suitable to close the section of edible films. It should come at the beginning of the section. Moreover, the paragraph before this one is too long and only reproduces the main findings of the studies, without discussing them. They should come summarized in a table. The same commentaries are all valid for section 9 (too long, does not critically say anything, the closing paragraph is confusing and in the wrong place).

Answer- Changes has been done and table has been added as per the suggestion

Round 2

Reviewer 1 Report

I think that the readability of the manuscript has  improved with the modifications and with the inclusion of tables, as suggested by other referee.

In my opinion, the paper can be published after minor corrections that I include in the following:

#1 Line 1266. I would eliminate the word nanoparticles in the sentence “… is classified as nanospheres or nanoparticles”.

The word “nanoparticles” can have different meanings but I think that in the paper it should be used in the wide sense of “nano-dimensional material (discontinuous phase)” of the nanocomposite, as stated by authors in sentence in line 1264. This nano-dimensional material can have different geometries, one of them beeing nanosphere. That is to say, is better to use the word “nanoparticle” for any type of nanofiller and the word “nanosphere” for spherical nanoparticles.

#2 In some places over the manuscript units, as for example, g/m2.h suscripts should be used for example Line 1370).

#3 Line 1433. “7 mm” should be written instead of “7mm” (an empty space should be added between the number and the unit”. This has to be corrected in some other lines.

Reviewer 3 Report

The authors improve a lot the quality of the article with the reviewer's suggestions. Therefore, I am willing to review my decision and now I recommend the publication of the revised manuscript.